



**Export fluxes of dissolved inorganic carbon to the Northern Indian Ocean**
**from the Indian monsoonal rivers**
Moturi S. Krishna[1], Rongali Viswanadham[1], Mamidala H. K. Prasad[1], Vuravakonda R. Kumari[1],
Vedula V. S. S. Sarma[1]
[1]CSIR-National Institute of Oceanography, Regional Centre, Visakhapatnam, 530017, India
*Correspondence to:* M. S. Krishna (moturi@nio.org)
**Abstract.** Rivers are strong source of dissolved inorganic carbon (DIC) to the adjacent coastal
waters. In order to identify the major sources of DIC in the Indian monsoonal estuaries and their
export flux to the north Indian Ocean, 27 major and medium estuaries along the Indian coast
were sampled during discharge period. An order of magnitude variability in DIC concentrations
was found within the Indian estuaries sampled (3.4 - 44.1mg l$^{-1}$) due to significant variability in
the size of rivers, precipitation pattern and lithology in the catchments. Dilution with high
precipitation (2500±500 mm) and exchange with ground waters of low DIC resulted in very low
concentrations of DIC in estuaries located in the southwest of India (6.6±2.1 mg l$^{-1}$) than the
estuaries located in the southeast (36.3±6.3 mg l$^{-1}$), northwest (30.3±8.9 mg l$^{-1}$) and northeast
(19.5±6.2 mg l$^{-1}$) regions of India. Though the range of stable carbon isotopes of DIC ($\delta^{13}C_{DIC}$)
indicates that DIC is largely contributed by weathering of silicate and carbonate minerals,
however, the storage of water in dams/reservoirs and intrusion of marine waters caused the
enrichment in stable carbon isotopic composition of DIC ($\delta^{13}C_{DIC}$). It is estimated that the Indian
monsoonal estuaries annually export ~10.4 Tg (1Tg=10$^{12}$ g) of DIC to the northern Indian
Ocean, of which the major fraction (74.2%) enters into the Bay of Bengal and the remaining
reaches to the Arabian Sea. It is mainly due to the fact that the Bay of Bengal receives ~378 km$^3$
yr$^{-1}$ of freshwater from the catchment area of about 0.96 million km$^{-2}$, whereas the Arabian Sea
receives only 122 km$^3$ yr$^{-1}$ of freshwater from the catchment area of only 0.23 million km$^2$.



Though the discharge from the Indian monsoonal rivers account for only 1.3% of global freshwater discharge, they disproportionately export 2.5% of the total DIC export by the world major rivers and 9.4% of the Asian rivers to oceans. The yield of DIC was found to be higher in the SW estuaries ($10.8 \pm 6.6$ g m$^{-2}$ yr$^{-1}$) than the other estuaries though they export only 0.3 Tg yr$^{-1}$ of DIC, which is more than an order of magnitude lower than the export by the NE (4.2 Tg yr$^{-1}$) and SE estuaries (3.5 Tg yr$^{-1}$), due to intense precipitation, favorable natural vegetation and tropical wet climate, high soil organic carbon and dominance of red loamy soils in catchments of the SW rivers. This study, therefore, reveals that significant variability in the lithology and hydrological and environmental conditions over the catchments strongly controls the concentrations and yield of DIC from the Indian monsoonal estuaries.

*Keywords:* dissolved inorganic carbon, export flux, Indian rivers, Bay of Bengal, Arabian Sea, North Indian Ocean

**1. Introduction**

Dissolved inorganic carbon (DIC) is the major constituent of carbon species and accounts for ~38% of the total fluvial carbon transport to the global oceans (Meybeck, 1993; Cai, 2011; Jarvie et al., 2017). World major river systems export annually about 33-400 Tg (1Tg=10$^{12}$g) of DIC to the global oceans (Ludwig et al., 1998; Mackenzie et al., 2004; Lerman et al., 2007). Chemical weathering of carbonate and silicate rocks and soils, and exchange with the ground water in the basin are the major sources of DIC into rivers (Meybeck, 1987; Gaillardet et al., 1999, Dessert et al., 2001; Viers et al., 2007; Raymond et al., 2008; Tamooh et al., 2013), besides in-stream processes, such as oxidation of organic carbon by heterotrophic bacteria (Mayogra et al., 2005; Battin et al., 2008; Hotchkiss et al., 2015; Samantha et al. 2015; Zou, 2016) and dissolution of atmospheric carbon dioxide ($CO_2$). Weathering of carbonate and



silicate rocks in the catchment, and uptake of DIC by aquatic plants and algae during
photosynthesis reaction in rivers are the sinks of the atmospheric $CO_2$ (e.g. Berner et al., 1983;
Raymond et al., 2008), while the oxidation of organic carbon is the source of $CO_2$ to the
atmosphere. DIC in rivers and estuaries is therefore strongly linked to the carbon cycle.
However, due to human interferences, DIC fluxes from the world major rivers have been found
to increase dramatically in the last century, for example, Mississippi (Cai, 2003; Raymond and
Cole, 2003; Raymond et al., 2008; Ren et al., 2015).  It has been noted that substantial alterations
in DIC lateral transport occurred from land to sea after the industrialization (Regnier et al., 2013;
Bauer et al., 2013).  The increase in riverine DIC flux was reported to have a significant impact
on the chemical composition (Williamson et al., 1994; Raymond and Cole, 2003; Findlay, 2010;
Tank et al., 2010) and carbon budget in the coastal waters (Cole et al., 2007; Dhillon and
Inamdar, 2013). Thus, identification of major sources of DIC and its riverine export flux
estimates to the coastal oceans are important for better understanding the carbon cycling and its
budget on both regional and global scales (Campeau et al., 2017).

Fluvial carbon fluxes from rivers in the tropical region ($30^o$ N to $30^o$S) is critical for

global carbon budgets because they contribute significant fraction of global DIC (48-64%),
freshwater discharge (66.2%) and suspended sediment load (73.2%) to the world oceans, despite
they occupy only ~43% of world's land area (Huang et al., 2012). Further, humid tropical
climate in the region supports the export of more fluvial carbon fluxes from the continental land
masses than the other climates in the world (Meybeck 1993; Ludwig et al., 1998).  However,
DIC fluxes from rivers in this region were not included in estimating the fluvial carbon fluxes to
global oceans due to the paucity of data.



Numerous studies have been documented on DIC export flux from the world major
rivers, for example, Mississippi (Raymond and Cole, 2003; Raymond et al., 2008; Cai et al.,
2008), Changjiang and Pearl (Cai et al., 2008), Congo (Wang et al., 2013) and large river
systems in the world (e.g. Gaillardet et al., 1999; Raymond et al., 2013).   Although some
measurements were carried out on DIC in the Indian estuaries, for example, Mandovi and Zuari
(Sarma et al., 2001), Godavari estuary (Sarma et al., 2011), Cochin (Gupta et al., 2009; Bhavya
et al., 2016), Hooghly (Mukhopadhyay et al., 2002; Samanta et al., 2015), Mahanadi (Pattanaik
et al., 2017) and Indian estuaries (Sarma et al., 2012), however, they focused only on air-water
exchange of $CO_2$.  Nevertheless, no estimations on DIC export fluxes to the north Indian Ocean
from the Indian subcontinent have been made so far.  For the first time, we made an effort here
to identify the major sources of DIC in the Indian monsoonal estuaries and to estimate their
export fluxes to the north Indian Ocean. The main objectives of this study are to (i) identify the
major sources and (ii) examine the potential reasons responsible for variability in concentrations
of DIC in the Indian monsoonal estuaries during the discharge (wet) period, and (iii) estimate the
DIC export fluxes to the north Indian Ocean by the Indian monsoonal rivers.
**2. Study region and Sampling**
**2.1 Study Area**
The Indian peninsula bifurcate the north Indian Ocean into the Bay of Bengal and the
Arabian Sea.  Although these two basins occupies the same latitudinal belt, their oceanographic
processes were reported to be remarkably different and attributed to significant differences in the
freshwater influx and associated physical and biological changes (Gauns et al., 2005). This is
because the glacial and peninsular rivers transport $1.63 \times 10^{12}$ $m^3$ $yr^{-1}$ of freshwater to the Bay of
Bengal (Subramanian, 1993) whereas only $0.3 \times 10^{12}$ $m^3$ $yr^{-1}$ to the Arabian Sea.   The large



freshwater influx leads to formation of a strong vertical salinity stratification (Varkey et al.,
1996), which results in the suppression of vertical mixing of nutrient rich sub-surface water with
that of surface, makes the Bay of Bengal relatively less productive (Prasannakumar et al., 2002)
than the Arabian Sea, which is one of the highly productive zones in the world (Madhupratap et
al., 1996; Smith, 2001; Barber et al., 2001) due to injection of nutrients into surface through the
seasonal upwelling and convective mixing (Shetye et al., 1994; Madhupratap et al., 1996;
Muraleedharan and Prasannakumar, 1996).

Discharge from the Indian peninsular rivers is fed by the monsoon induced precipitation

over the Indian subcontinent, which receives >80% of its annual rainfall during the southwest
(SW) monsoon period (June-September) (Soman and Kumar, 1990). Though some amount of
rainfall occurs during the NE monsoon (December-March), it will not generate discharge as it
will be stored in dams and reservoirs for domestic, industrial and irrigation purposes. Discharge
from the Indian peninsular rivers is therefore occurs only during the SW monsoon season (Vijith
et al., 2009; Sridevi et al., 2015) and hence, termed as monsoonal rivers. Since the freshwater
discharge from the Indian monsoonal rivers is limited to only few months (June – October) in a
year, unlike the European and American rivers, the entire estuary may be filled with freshwater
without any vertical salinity gradient (Vijith et al., 2009; Sridevi et al., 2015) during this period.
As virtually there is no discharge during the dry period, the discharge during SW monsoon (wet
period) is equivalent to the annual discharge from the monsoonal rivers. Based on the rainfall
intensity, forest cover, vegetation and soil type in the catchment, estuaries sampled in the present
study were  categorized into 4 groups, namely the northeast (NE), southeast (SE), southwest
(SW) and northwest (NW) estuaries of India (Fig. 1).  The SW region of India is characterized
by the intense rainfall during the SW monsoon (~3000 mm) following the NE (1000-2500 mm),




SE (300-500 mm) and NW (200-500 mm) regions of India (Soman and Kumar, 1990). The SW
rivers drain red loamy soils while the NW rivers drain black soils.  The rivers reaching the Bay
of Bengal (NE and SE estuaries) drain the red loamy and alluvial soils in their upper and lower
catchments respectively, except the major rivers Godavari and Krishna, which also drain black
soils in their upper catchment along with red loamy and alluvial soils in their middle and lower
catchments (Geological Survey of India; www. gsi.gov.in). Based on the discharge, monsoonal
estuaries in this study were divided into two types, namely, the minor ($<150 \text{ m}^3\text{s}^{-1}$) and major
($>150 \text{ m}^3 \text{ s}^{-1}$) estuaries.

**2.2 Sample collection**

Estuaries are known to be biologically active spots in the aquatic ecosystem and therefore

significant modification of DIC (through autotrophic primary production or heterotrophic
respiration) is possible.  Hence, samples were collected from mouth of the estuaries rather than
from mid or upstream rivers for reliable export fluxes of DIC to the coastal ocean. Further, to
minimize the inter-annual variability in DIC concentrations, sampling was conducted in
discharge period of two years, i.e., 2011 and 2014 and the mean DIC concentration in each
estuary was used for export flux estimations.  Each estuary was sampled at 3 to 5 locations
between the upstream river (near zero salinity) and mouth of the estuary in order to minimize the
spatial variability in DIC concentrations, and the mean concentrations were used for flux
estimates.   Further, samples were collected in mid-stream of the estuary using a local
mechanized boat to avoid the contamination from banks.

In-situ measurements and sample collection was done in the 27 estuaries (Fig. 1) during

the SW monsoon season of the years, 2011 and 2014. Surface water samples at each location
were collected for phytoplankton biomass (Chl-*a*), DIC and dissolved oxygen (DO).  Samples



for DIC were collected in air-tight crimp-top glass bottles and added poison (mercuric chloride)
to arrest the biological activity.  DO analysis was carried out at a temporary shore laboratory set
up for sample processing after the completion of sampling on each day.  Water samples were
filtered through GF/F (nominal pore size: 0.7μm) under moderate vacuum and stored in liquid
nitrogen for Chl-*a* analysis at the NIO.
**3. Methods**

Temperature and salinity at the sampling locations were measured using a CTD system

(Sea Bird Electronics, SBE 19 plus, United States of America). Concentrations of DO were
determined by Winkler's method (Carritt and Carpenter, 1966) using an auto titrator (Metrohm,
Switzerland) with potentiometric end point detection.  The analytical precision of the method
was ±0.07% (RSD). DIC concentrations in water samples were measured at our Institute
laboratory using Coulometer (UIC Inc., USA) connected to an automatic sub-sampling system.
Based on the repeated analysis of samples and standards, the precision of the method was ±1.8
μmol l$^{-1}$.  The certified reference materials (CRM) supplied by Dr. A.G. Dickson, Scripts
Institute of Oceanography, USA and internal standards were used to test the accuracy of our DIC
measurements and it was found to be within ± 0.2 to 0.3%.  Chlorophyll-*a* (Chl-*a*) on the filter
was extracted into di-methyl formamide (DMF) and measured the extract fluorometrically using
a spectrofluorophotometer (Varian Eclipse, Varian Electronics., UK) following Suzuki and
Ishimaru (1990).  Annual mean discharge data of rivers was taken from Meybeck and Ragu
(1995, 1996), Central Water Commission, New Delhi (2006, 2012) and Kumar et al. (2005). Soil
organic carbon data was taken from Kishwan et al. (2009) and rainfall data was obtained from
Soman and Kumar (1990). Dissolved organic carbon (DOC) data for the Indian estuaries was
taken from Krishna et al. (2015).





Total export flux of DIC from each river was estimated by multiplying the mean
concentrations of DIC in an estuary with the mean annual discharge. Spatial variability in DIC
concentrations in estuaries was minimized to a large extent by collecting samples from head to
mouth of the estuary while the inter-annual variability by collecting samples during discharge
periods of two years. However, variability in DIC concentrations within the discharge period
results in some uncertainties in our estimations of DIC export fluxes. Time series measurements
in the Godavari estuary (our unpublished results) revealed that the variability in DIC
concentrations within the discharge period is up to 10%. Therefore, the error associated with our
DIC flux estimates can be about ±10%. DIC fluxes normalized by catchment area (yield) were
calculated by dividing the total DIC export flux of the river by its catchment area.
**3. Results**
Prevailing hydrographic conditions in Indian estuaries during the sample collection were given in
detail elsewhere (Sarma et al., 2012, 2014; Krishna et al., 2015). Briefly, mentioned here for
ready reference. Surface water temperature was found to be higher in estuaries located on the
east coast (mean 30.86±1.23°C) than the west coast (27.32±1.49°C) of India. Salinity varied
broadly from near zero (0.06) to 28.78 during the study period. Relatively higher salinities (>20)
were recorded by the medium estuaries, which receives relatively lower freshwater discharge
from the upstream river, for example, Nagavali (28.78), Vaigai (24.63) and Rushikulya (20.70).
Dissolved oxygen saturation varied from as low as 62.6% to as high as 105%, with a mean
saturation of 89.9±11.4% in the estuaries sampled. Chlorophyll-*a* (Chl-*a*) concentrations varied
broadly from 0.8 to 10.7 mg m$^{-3}$, with relatively higher mean concentrations in the SE (4.7 mg
m$^{-3}$) followed by the SW (3.0 mg m$^{-3}$) estuaries. However, relatively low Chl-*a* was observed in
the medium (2.6±1.3 mg m$^{-3}$) than in the major estuaries (3.2±2.1 mg m$^{-3}$).



### 3.1 Concentrations and $\delta^{13}$C of DIC ($\delta^{13}C_{DIC}$) in the Indian monsoonal estuaries

DIC concentrations in Indian estuaries widely varied from 3.4 (Bharathappuzha) to 44.1 mg l$^{-1}$ (Vellar), with a significant spatial variability (Fig. 2). More than five times higher mean concentrations were observed in the SE (36.3±6.3 mg l$^{-1}$) and NW estuaries (30.3±8.9 mg l$^{-1}$) than in the SW estuaries (6.6±2.1 mg l$^{-1}$), and intermediate concentrations were found in the NE estuaries (19.5±6.2 mg l$^{-1}$). DIC concentrations were found to be similar (homoscedastic Student's t-test; p=0.76) in the major (22.7±13.6 mg l$^{-1}$) and medium (21.1±13.2 mg l$^{-1}$) estuaries. The $\delta^{13}C_{DIC}$ varied from -13.0 to 2.5‰, with a significant spatial variability (Fig. 3) in the estuaries sampled. Relatively depleted values were observed in the west flowing estuaries of NW (-11.1±2.3‰) and SW (-7.4±1.9‰) than the east flowing estuaries of NE (-3.5±2.8‰) and SE (-2.7±5.2‰) regions of India.

Annual export flux of DIC from the individual estuaries to coastal ocean varied broadly from 0.009 Tg (Chalakudi) to as high as 2.32 Tg (Krishna). Annually, the NE estuaries export higher DIC flux of 4.21 Tg followed by the SE (3.50 Tg) and NW estuaries (2.38 Tg). Whereas, the SW estuaries recorded the lowest export flux of 0.30 Tg which is an order of magnitude lower than the export flux by the NE and SE estuaries (Fig. 2). The Indian monsoonal estuaries together export about 10.4 Tg yr$^{-1}$ of DIC to the northern Indian Ocean, of which 7.7 Tg (74.2%) enters into the Bay of Bengal and the remaining into the Arabian Sea (2.7 Tg). The estuaries, Krishna (2.32 Tg), Godavari (1.45 Tg) and Haldia (1.16 Tg) together responsible for the transport of 64% of total riverine DIC export to the Bay of Bengal by the Indian monsoonal rivers. The yield of DIC ranged from 2.7 (Bharathappuzha) to 21.6 g m$^{-2}$ yr$^{-1}$ (Mandovi), excluding the exceptionally high yield of 113.4 g m$^{-2}$ yr$^{-1}$ from Haldia estuary. The west flowing rivers to the Arabian Sea are characterized by relatively higher yield of DIC (mean 10.4±5.6 g m$^{-}$





$^{2}$ yr$^{-1}$) than the east flowing rivers to the Bay of Bengal (7.3±4.6 g m$^{-2}$ yr$^{-1}$). Among the estuaries
sampled, the SW and SE estuaries recorded higher (10.8±6.6g m$^{-2}$ yr$^{-1}$) and lower (5.8±2.3g m$^{-2}$
yr$^{-1}$) yields of DIC respectively. The NW (9.5±4.0 g m$^{-2}$ yr$^{-1}$) and NE (8.6±5.7g m$^{-2}$ yr$^{-1}$)
estuaries recorded intermediate values.

### 4. Discussion

Hydrographic characteristics of the Indian monsoonal estuaries during the study
(discharge) period were described elsewhere (Sarma et al., 2012, 2014; Krishna et al., 2015).
Strong flow from the upstream rivers due to the SW monsoon-induced precipitation over the
catchment makes most of the estuaries less saline (near zero), except the minor estuaries,
Nagavali, Vaigai and Rushikulya, during the study period. No vertical salinity stratification was
observed in estuaries during the study period, consistent with earlier observations in the Indian
estuaries during discharge period (Vijit et al., 2009; Sridevi et al., 2015), unlike the European
and American estuaries (Christopher et al., 2002).

### 4.1 Variability of DIC concentrations in the Indian monsoonal estuaries

Mean DIC concentration found in this study (21.9±13.2 mg l$^{-1}$; range: 3.4 to 44.1 mg l$^{-1}$)
is similar to those observed earlier in the Indian estuaries, for example, Ganga-Brahmaputra (23
mg l$^{-1}$; Singh et al., 2005), Hooghly (21.8 mg l$^{-1}$; Samanta et al., 2015) and Mahanadi (15.0;
Pattanaik et al., 2017) and elsewhere in the world, for instance, York river estuary (6-21 mg l$^{-1}$;
Raymond and Bauer, 2000), Yangtze river (28 mg l$^{-1}$; Cai et al., 2008), British rivers (median 4-
43 mg l$^{-1}$; Jarvie et al., 2017), Seri, central Japan (17.6-21.9 mg l$^{-1}$; Ishikawa et al., 2015), the
Red river, Vietnam (9.1-29.9 mg l$^{-1}$; Quynh et al., 2016) and Xi river, southwest China (18-
45.6mg l$^{-1}$, Zou, 2016).  However, mean DIC concentrations in the Indian estuaries (21.9±13.2





mg l$^{-1}$) are higher than the global mean of 10.3 mg l$^{-1}$ (Meybeck and Vorosmarty, 1999) and the
Asian rivers (12.7 mg l$^{-1}$) in the tropical region (30°N-30°S; Huang et al., 2012), but lower than
the rivers draining into the Gulf of Trieste (N Adriatic) (37-66 mg l$^{-1}$; Tamse et al., 2014).

Among the estuaries sampled along the Indian coast, the SW estuaries are characterized

by significantly lower mean concentrations of DIC (6.6±2.1 mg l$^{-1}$) than the SE (36.3±6.3 mg l$^{-1}$)
), NE (19.5±6.2 mg l$^{-1}$) and NW (30.3±8.9 mg l$^{-1}$) estuaries. DIC concentrations in estuaries are
mainly governed by the hydrological (precipitation and runoff), lithological (type and dominance
of rocks and soils) and environmental (temperature, climate and vegetation) conditions, and
anthropogenic activities (deforestation and land use change) in the catchment, and in-stream
physical and biological processes such as exchange with ground water (Finlay, 2003; Shin et al.,
2011; Maher et al., 2013) and atmospheric $CO_2$, autotrophic production and heterotrophic
decomposition of organic matter (McConnaughey et al., 1994; Abril et al., 2003; Finlay and
Kendall, 2007). Since many of these processes are largely dependent on the size of the river and
its catchment, the lower DIC concentrations in the SW estuaries of this study could be due to the
size of the rivers. This is because, the SW rivers are small both in terms of discharge (46 km$^3$ yr$^{-1}$
) and catchment area (total catchment area: 0.02 M km$^2$) than the SE (102 km$^3$ yr$^{-1}$ and 0.45 M
km$^2$, respectively), NE (276 km$^3$ yr$^{-1}$ and 0.53 M km$^2$) and NW (75 km$^3$ yr$^{-1}$ and 0.21 M km$^2$)
rivers. However, DIC concentrations showed significant positive relationship with catchment
area (r$^2$=0.75; p<0.001; Fig. 4a) and negative relationship with volume of discharge (r$^2$=0.57;
p<0.001; Fig. 4b) only in the medium estuaries (discharge: <150 m$^3$s$^{-1}$), suggesting that an area
of catchment and magnitude of discharge controls DIC concentrations largely in the medium
estuaries rather than the major estuaries.





Indian monsoonal estuaries were reported as a source of $CO_2$ to the atmosphere during
discharge period (Sarma et al., 2001, 2011, 2012; Gupta et al., 2009; Bhavya et al., 2016) due to
the microbial decomposition of terrestrial organic matter brought by the rivers. This suggests
that the DIC input from dissolution of atmospheric $CO_2$ in estuaries can be ruled out, however,
heterotrophic decomposition of organic matter adds significant amount of DIC to the Indian
estuaries during discharge period. A fairly good positive correlation between DIC and DOC
concentrations ($r^2=0.34$, $p<0.01$), except few medium estuaries, suggests that DIC addition
through microbial degradation of particulate organic matter is significant in the Indian estuaries.
Except the NW estuaries, which recorded relatively depleted $\delta^{13}C$ of DIC ($\delta^{13}C_{DIC}$), the positive
correlation between $\delta^{13}C_{DIC}$ and DOC concentrations ($r^2=0.35$, $p<0.01$), as was observed
elsewhere (Xi river, Zou et al., 2016), confirms that oxidation of organic matter is one of the
main DIC sources in the Indian estuaries. On the other hand, autotrophic production removes
DIC as it converts DIC to organic carbon. Significant negative correlation between chlorophyll-
$a$ and DIC concentrations ($r^2=0.47$, $p<0.01$), except few SE estuaries where elevated
phytoplankton biomass (Chl-a: $>5$ mg m$^{-3}$) was recorded, suggesting that autotrophic removal of
DIC is also significant in the Indian monsoonal estuaries during the study period. Significance
of DIC addition by heterotrophic decomposition and removal by autotrophic production in the
Indian estuaries was confirmed by a fairly good positive correlation between $\delta^{13}C_{DIC}$ and
dissolved oxygen saturation ($r^2=0.49$, $p<0.01$), (depleted $\delta^{13}C_{DIC}$ values at low % of DO
saturation), except NW estuaries, which recorded depleted $\delta^{13}C_{DIC}$ ($<-10.0‰$). This is because
the microbial decomposition of organic matter results in depleted $\delta^{13}C_{DIC}$ due to preferential
release of $^{12}C$ over $^{13}C$ in to DIC pool while removal of DIC by autotrophic production enriches
the residual DIC due to preferential uptake of $^{12}C$ over $^{13}C$ during photosynthesis reaction.




As found in many estuaries over the world, submarine ground water exchange strongly
influences DIC concentrations in Indian estuaries, for example, Rengarajan and Sarma (2015)
found 3 to 4 times higher DIC concentrations in the ground water compared the estuarine waters
of the Godavari and estimated that submarine ground water discharge contributes up to 52% of
DIC concentrations in the Godavari estuarine system.  The measured DIC concentrations in
ground waters along the entire Indian coast (Dr. BSK Kumar, personal communication) showed
strong spatial variability with relatively lower concentrations in the SW (mean $32\pm19$ mg l$^{-1}$)
than the SE ($106\pm56$), NE ($92\pm31$) and NW ($84\pm54$ mg l$^{-1}$) regions of India during discharge
period.  Though the DIC concentrations in ground waters were higher by about 3 to 5 times than
the concentrations found in the Indian estuaries, however, exchange of ground water with
relatively low DIC concentrations in the SW region could have, at least partly, caused the lower
DIC concentrations in the SW estuaries.
Spatial distribution of bedrock and soils over the Indian subcontinent shows that
Narmada and Tapti rivers and upper reaches of Godavari and Krishna rivers drain the igneous
rocks (Deccan traps) while the other rivers flow through the metamorphic rocks (Pre-Cambrian),
the predominant rock type in south India.  However, Haldia and lower reaches of the SE rivers
drain the sedimentary rocks (Geological Survey of India, https://www.gsi.gov.in). Although, the
chemical weathering rates were reported to be higher for Deccan Trap basalts (Das et al., 2005;
Singh et al., 2005), however, higher DIC concentrations were also found in estuaries draining the
metamorphic rocks, suggesting that strong influence of factors other than the bedrocks in the
catchment.  Spatial distribution of soils shows that lateritic soils, which are poor in lime and
silicate, occupied the catchment of the SW rivers. Chemical weathering rates are relatively lower
in the lateritic than the non-lateritic soils and the consumption of atmospheric/soil $CO_2$ through



silicate weathering is lower by ~2 times in the former than the latter (Boeglin and Probst, 1998).
Though the upper reaches of the east flowing rivers (NE and SE) drain the lime-poor red and
yellow soils, however, they are dominated by the lime-rich alluvial soils in their lower reaches.
Upper reaches of Krishna and Godavari also drain the lime-rich black soils. The dominance of
lateritic soils, which are relatively less susceptible to chemical weathering than the non-lateritic
soils, over the catchments of the SW rivers could have, at least in part, lowered the DIC
concentrations in SW estuaries during the study period.

The SW region of India receives highest amount of rainfall during the SW monsoon

($2500 \pm 500$mm) than the SE ($400 \pm 50$), NE ($1000 \pm 200$) and NW ($750 \pm 250$mm) regions of India
(Soman and Kumar, 1990).  Though the intense rainfall in the SW region is expected to cause
higher weathering rates and therefore higher DIC (e.g., Gupta et al., 2011), the observed lower
DIC concentrations in the SW estuaries could be due to the dilution.  The catchment area
normalized volume of discharge was found to be higher in the SW estuaries ($1.71$ m$^3$ m$^{-2}$) than in
the SE ($0.17$), NE ($0.6$) and NW ($0.32$m$^3$ m$^{-2}$) estuaries, suggesting that significant dilution of
DIC concentrations in the SW estuaries.  A strong negative correlation between precipitation in
the catchment and DIC concentration in estuaries ($r^2 = 0.89$, $p < 0.001$; Fig. 5) confirms that DIC
concentration in Indian estuaries are rather controlled by the intensity of precipitation over the
catchment.
**4.2 $\delta^{13}$C of DIC in the Indian monsoonal estuaries**

The stable isotopic composition of DIC ($\delta^{13}$C$_{DIC}$) is a well-established and widely used

tracer to identify the major sources of DIC in rivers (e.g. Singh et al., 2005; Tamooh et al., 2013;
Samanta et al., 2015; Zou, 2016) because each of the DIC sources have a distinct $\delta^{13}$C$_{DIC}$ ratios
(Deines et al., 1974).  DIC originated by dissolution of atmospheric CO$_2$ is about -7 to -8‰



(Coplen et al., 2002) whereas it is about -26 to -27‰ if DIC is derived from oxidation of organic
matter produced by $C_3$ plants (O'Leary, 1988). The $\delta^{13}C$ of DIC generated by soil $CO_2$ dissolved
carbonic acid weathering of silicates is about -17 to -21‰ (Solomon and Cerling, 1987) while it
is close to -9‰ for carbonate rocks because half of the carbon comes from carbonate rocks (0‰,
Land, 1980) during weathering.  Whereas, the weathering of carbonate and silicate minerals
yield $\delta^{13}C_{DIC}$ values -7 to -8‰ and -3 to -4‰, respectively, if the carbonic acid formed by the
dissolution of atmospheric $CO_2$. Although, DIC derived from different sources have distinctly
different $\delta^{13}C_{DIC}$ values, however, the interpretation the $\delta^{13}C_{DIC}$ values for identification of its
sources is still challenging (Amiotte-Suchet et al., 1999; Campeau et al., 2017) due to the
isotopic fractionations associated with complex mixture of sources and processes such as
photosynthesis (O'Leary, 1988; Finlay, 2004; Parker et al., 2005, 2010), respiration (Finlay,
2003; Waldron et al., 2007), DOC photo-oxidation (Opsahl and Zepp, 2001; Vahatalo and
Wetzel, 2008), anaerobic metabolism (Waldron et al., 1999; Maher et al., 2015) and equilibration
with atmospheric $CO_2$.

The range of $\delta^{13}C_{DIC}$ found in this study (-13.0 to 2.5‰) was similar to those reported

earlier in various rivers, for example, Brahmaputra (Singh et al., 2005), Rhine (Buhl et al., 1991),
Ottawa (Telmer et al., 1999), St. Lawrence (Yang et al., 1996), Nanpan and Beipan rivers,
southwest China (Zou, 2016) and Tana river, Kenya (Tamooh et al., 2013). The range of $\delta^{13}C_{DIC}$
in this study indicates a variety of sources, including silicate and carbonate weathering and
marine waters, contributes DIC to the Indian monsoonal estuaries during the study period.
Relatively depleted $\delta^{13}C_{DIC}$ in the west flowing river estuaries of NW (mean -11.1±2.3‰) and
SW (mean: -7.4±1.9‰) regions suggest that DIC is contributed from silicate and carbonate
weathering by the carbonic acid, produced from the dissolution of both soil $CO_2$ and atmospheric



$CO_2$. Zou (2016) found the $\delta^{13}C_{DIC}$ values in the range of -13.9 to -8.1 ‰ in the Nanpan and
Beipan rivers of SW China and were attributed to dominant contribution of DIC from weathering
of carbonate minerals.  Relatively enriched $\delta^{13}C_{DIC}$ in the east flowing river estuaries of NE (-6.5
to 0.7; mean: -3.5±2.8‰) and SE (-7.9 to 2.5‰; -2.7±5.2‰) indicates that major contribution of
DIC is from chemical weathering of carbonate rocks by atmospheric $CO_2$ dissolved carbonic
acid or acid from non-carbon sources (Li et al., 2008). Weathering of carbonate minerals by acid
sources other than carbonic acid causes enrichment compared to weathering by carbonic acid due
to lack of contribution from $\delta^{13}C$-depleted  carbonic acid of soil $CO_2$ (-17 to -21‰) or
atmospheric $CO_2$ (-7 to -8‰) origin to the $\delta^{13}C$-enriched carbonate rocks (0‰, Land 1980).

In addition to the sources, hydrological and biological processes also influence the

$\delta^{13}C_{DIC}$ in streams/rivers. For example, heavy precipitation in the SW region (2500±500mm)
than the other regions tends to cause depletion in $\delta^{13}C_{DIC}$ values due to shorter residence time of
soil water (Amiotte-Suchet et al., 1999) while $CO_2$ out gassing causes enrichment due to
accumulation of $^{13}C$ during diffusive efflux (Clark and Fritz, 1997) in stored water bodies.  Many
of the east flowing rivers are major and are dammed at many locations (e.g. Godavari, Krishna
and Cauvery) for domestic, industrial and irrigation purposes. $CO_2$ out gassing due to
heterotrophic decomposition of organic matter and autotrophic production significantly alters the
$\delta^{13}C_{DIC}$ signatures in reservoirs (Shin et al., 2001). Further, equilibrium with atmospheric $CO_2$ in
the reservoirs due to no/lean flow leads to enrichment in the $\delta^{13}C_{DIC}$ values (Brunet et al., 2005;
Bouvillion et al., 2009; Zeng et al., 2011; Tamooh et al., 2013).  Hence, relatively enriched
$\delta^{13}C_{DIC}$ in the NE and SE estuaries could also be due to the storage of water in reservoirs/dams.
A significant positive correlation between DIC concentrations and $\delta^{13}C_{DIC}$ ($r^2$=0.77; p<0.001;
Fig. 7), excluding the positive values, indicate that significant contribution of DIC from





oxidation of particulate organic carbon in dams/reservoirs or stored water bodies. Shin et al.
(2011) attributed the stream $\delta^{13}C_{DIC}$ values of -6.9±1.6‰ and -7.8±1.5‰ in silicate and
carbonate dominated catchments, respectively, in tributaries of the Han River, South Korea to
$CO_2$ out gassing. Positive $\delta^{13}C_{DIC}$ values (>0‰) were observed only in Rushikulya (0.1‰),
Nagavali (0.7‰) and Vaigai (2.5‰) in which relatively higher salinities (>20) were found
during the study period. This is concurrent with earlier observations in the Indian estuaries,
Hooghly (Samanta et al., 2015) and Cochin (Bhavya et al., 2016) where relatively enriched
$\delta^{13}C_{DIC}$ were found at higher salinities. A strong positive correlation was found between $\delta^{13}C_{DIC}$
and salinity (Fig. 6; $r^2$=0.71, p<0.001), suggesting that $\delta^{13}C_{DIC}$ values in the Indian estuaries are
influenced by the intrusion of marine waters ($\delta^{13}C_{DIC}$: -1 to 2‰).
**4.3 Total DIC export by the Indian monsoonal rivers to the north Indian Ocean**
Indian monsoonal rivers annually export ~10.4 Tg of DIC to the north Indian Ocean.
Nearly three fourth of this amount (7.7 Tg) reaches to the Bay of Bengal while the remaining
into the Arabian Sea. It is mainly attributed to the magnitude of discharge because the Bay of
Bengal annually receives 378 $km^3$ of freshwater from the catchment area of about 0.96 M $km^{-2}$,
whereas the Arabian Sea receives only 122 $km^3$ of freshwater from the catchment area of only
0.23 M $km^2$. Although the increase in volume of discharge dilutes the DIC flux from rivers
(Jarvie et al., 1997; Shanley et al., 2002), bicarbonate fluxes to the Gulf of Mexico were reported
to increase with the volume of discharge from the Mississippi river (Raymond and Oh, 2007) due
to small dilution factor.
The total DIC export by the Indian monsoonal estuaries (10.4 Tg $yr^{-1}$) is only 2.5% of the
total DIC export by the world major rivers (400 Tg $yr^{-1}$), and 9.4% of the export by the Asian



rivers (111Tg yr$^{-1}$; Huang et al., 2012). The DIC export from the Indian estuaries is far less than
the DIC export by the American (61.4 Tg yr$^{-1}$) and African (17.7 Tg yr$^{-1}$) rivers and major rivers
draining to the tropical Atlantic from South America and Africa (53Tg yr$^{-1}$, Araujo et al. 2014).
It is mainly due to the fact that the volume of discharge from the Indian monsoonal rivers is very
low (~500 km$^3$) compared to the American (11,799 km$^3$) and African (3,786 km$^3$) rivers.
However, the Indian monsoonal rivers are exporting DIC disproportionately to the north Indian
Ocean because they account for only 1.3% of the global river discharge but export 2.5% of the
global riverine DIC to the oceans. Disproportionate DIC fluxes from the tropical regions are
mainly attributed to the favourable climatic conditions, lithology and land use cover (Huang et
al., 2012) in this region. Relatively higher export fluxes from the Indian rivers could be due to
higher weathering rates of silicate and carbonate minerals in the drainage basins of the Indian
rivers (Das et al., 2005; Gurumurty et al., 2012; Pattanaik et al., 2013)

Krishna et al. (2015) reported that Indian monsoonal estuaries export 2.32 Tg yr$^{-1}$ of

dissolved organic carbon (DOC) to the north Indian Ocean. When combined the total fluvial
dissolved carbon flux would be 12.71 Tg yr$^{-1}$. This indicate that the total fluvial dissolved
carbon export to the north Indian Ocean by the Indian monsoonal estuaries is predominantly
contributed by DIC (~81%) than DOC, consistent with earlier reports elsewhere in the world, for
example, the British rivers (80%, Jarvie et al., 2017). Since the catchment area of the Indian
monsoonal rivers ranged widely from as low as 0.001 M km$^2$ to as high as 0.313 M km$^2$, the
export fluxes of DIC were normalized with the catchment area of river to obtain DIC yield from
each river in order to examine various factors controlling the lateral DIC export to the north
Indian Ocean.
**4.4 Yield of DIC from the Indian monsoonal rivers**





The yield of DIC found in this study (mean $8.7 \pm 5.2$ g m$^{-2}$ yr$^{-1}$) is similar those found
earlier in rivers from the tropical region (30$^o$N – 30$^o$S) of the Asian continent (9.79 g m$^{-2}$ yr$^{-1}$;
Huang et al., 2012), but significantly higher than the American (3.33 g m$^{-2}$ yr$^{-1}$) and African
(0.63 g m$^{-2}$ yr$^{-1}$) continents of this tropical region (Huang et al., 2012). The yield of DIC from
river catchment were reported to be controlled by the hydrological (precipitation, runoff and
groundwater exchange) and environmental (temperature, type and dominance of soils, soil
organic carbon, natural vegetation and forest cover) conditions and anthropogenic activities (land
use change and deforestation) in the catchment (Raymond et al., 2008; Huang et al., 2012).
Although the SW estuaries annually export relatively less DIC to the north Indian Ocean (0.30
Tg) due to their lower volume of discharge (46 km$^3$ yr$^{-1}$) from relatively smaller catchment area
(0.02 M km$^2$) than the SE (3.50 Tg, 102 km$^3$ yr$^{-1}$ and 0.43 M km$^2$ respectively), NE (4.21 Tg, 276
and 0.53) and NW (2.38 Tg, 75 km$^3$ yr$^{-1}$ and 0.21 M km$^2$) estuaries, strikingly, the higher yield of
DIC was found in the former ($10.8 \pm 6.6$ g m$^{-2}$ yr$^{-1}$) than the latter ($5.8 \pm 2.3$, $8.6 \pm 5.7$ and $9.5 \pm 3.9$ g
m$^{-2}$ yr$^{-1}$, respectively). This suggests that strong control of catchment and/or in-stream processes
on yield of DIC from the monsoonal rivers. However, DIC yield showed significant positive
correlation with the volume of discharge ($r^2=0.66$, $p<0.001$) in medium estuaries and no such
relationship was found in the major estuaries. Significant negative relationships were observed
between DIC yield and catchment area in the medium ($r^2=0.52$, $p<0.001$) and major estuaries
($r^2=0.49$, $p<0.001$).  This suggests that high precipitation over small catchments increases the
DIC yield from the Indian estuaries because the dense precipitation increases the scouring of
DIC from soils and rocks in their catchment. A strong linear relationship between the yield of
DIC and the intensity of precipitation ($r^2=0.64$, $p<0.001$ Fig. 8a) confirms that dense
precipitation increases the export yield of DIC.  This could be one the reasons for the observed

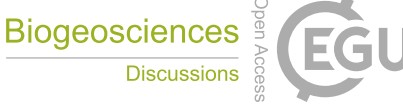

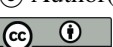

higher yield of DIC in the SW estuaries which receives high precipitation (2500±500mm) over
the small catchment area (0.02 M km$^2$).

Ground water exchange do not appears to be controlling DIC yield from the Indian

monsoonal estuaries because the groundwater DIC concentrations were lower in the SW (32±19
mg l$^{-1}$) than the other regions SE (106±56), NE (92±31) and NW (84±54mg l$^{-1}$). Existing natural
vegetation of tropical moist deciduous and tropical wet evergreen and semi evergreen forests in
the SW region also could have increased DIC yield from the SW estuaries compared to the other
estuaries as this vegetation favors the export fluxes of DIC. The drainage basins of the Indian
monsoonal estuaries are largely under the tropical dry and wet climate except the SW rivers,
Narmada and Tapti. The rivers Narmada and Tapti are under the arid and semiarid climate while
the SW rivers are under the tropical wet climate which was also reported to facilitate the riverine
export of material from drainage basin to the coastal ocean.

Catchments of the SW rivers are largely occupied by the cation deficient lateritic soils

and therefore precipitation of carbonate minerals in soils is poor. As a result, the soil inorganic
carbon content in surface (100cm) soils of the catchment of SW rivers was lower than in
catchments of the other monsoonal rivers studied (Sreenivas et al., 2016). On the other hand, the
authors (Sreenivas et al., 2016) and Krishwan et al. (2009) found that the soil organic carbon was
higher in the former than the latter. The relationship between soil inorganic and organic carbon is
primarily dependent on the soil characteristics in the catchment. For example, Guo et al. (2016)
demonstrated that increase in the soil organic carbon content enhanced the soil inorganic carbon
in the cropland of upper Yellow river delta, China. A strong positive relationship between soil
organic and inorganic carbon was also found in the Yanqi river basin, northwest China (Wang et
al., 2015), and soils in the America (Stevenson et al., 2005) and Canada (Landi et al., 2003). On



the other hand, a negative relationship was found between soil organic and inorganic carbon in
the North China Plain (Huang et al., 2006) and west Loess Plateau (Zeng et al., 2008). The
negative relationship is mainly due to the higher production of $CO_2$ by decomposition of soil
organic carbon and root respiration resulting in the formation of acidic conditions that lead to
dissolution of soil carbonates. The higher soil organic carbon in the catchment of the SW than in
catchment of the SE, NE and NW rivers (Kishwan et al., 2009; Sreenivas et al., 2016) therefore,
produces more $CO_2$ through microbial decomposition and causes  dissolution of  soil carbonates
leading to the higher yield of DIC from the SW estuaries.  A significant linear correlation
between soil organic carbon content and DIC yield in this study ($r^2$=0.65, p<0.001; Fig. 8b)
suggests that strong influence of soil organic carbon content in the catchment on DIC yield from
the Indian monsoonal rivers.  However, basin scale studies are required for comprehensive
understanding of the influence of environmental and anthropogenic factors on DIC export fluxes
from the Indian monsoonal rivers.
**5. Summary**

In order to examine the variability of dissolved inorganic carbon (DIC) concentrations

and to identify its major sources in the Indian monsoonal estuaries, and to estimate the riverine
export fluxes of DIC to the north Indian Ocean, we sampled a total of 27 major and medium
estuaries along the Indian coast during wet period.  An order of magnitude variability was found
in DIC concentrations among the estuaries sampled (3.4 - 44.1mg $l^{-1}$), with a lower mean
concentrations of 6.6±2.1 mg $l^{-1}$ in estuaries located in the SW region of India.  It is attributed to
significant variability in the size of rivers, precipitation pattern and lithology in their catchments.
Magnitude of discharge, catchment area and in-stream processes are appears to be important
factors for medium estuaries rather than major estuaries in controlling the concentration and



yield of DIC, probably due to a significant variability in lithology and hydro-geological and

environmental conditions in the catchments. Indian monsoonal estuaries annually export ~10.4

Tg of DIC to the north Indian Ocean, of which 7.7 Tg enters in to the Bay of Bengal while the

Arabian Sea receives only 2.7 Tg. It is mainly attributed to the volume of river discharge as

former receives ~378 km$^3$ yr$^{-1}$ while the latter receives only 122 km$^3$ yr$^{-1}$of freshwater from the

Indian monsoonal rivers. The range of $\delta^{13}C_{DIC}$ found in this study suggests that DIC is largely

contributed from weathering of silicate and carbonate minerals by carbonic acid formed by

dissolution of both soil and atmospheric $CO_2$. However, relatively enriched $\delta^{13}C_{DIC}$ in the east-

flowing river estuaries indicated the storage of water in dams/reservoirs and intrusion of marine

waters. Dense rainfall (2500±500mm) and higher soil organic carbon content (101.4 g ha$^{-1}$) in

the catchment of SW rivers than in the catchment of the other rivers resulted in higher yield of

DIC from the former than the latter.

**6. Acknowledgements**

We would like to thank the Director, CSIR - National Institute of Oceanography (NIO), Goa, and

the Scientist-In-Charge, NIO-Regional Centre, Visakhapatnam for their kind support and

encouragement. We also acknowledge Dr. M. Dileep Kumar, NIO, Goa for his guidance and

encouragement. The work is part of the Council of Scientific and Industrial Research (CSIR),

funded research project. This publication has NIO contribution number .....

**7. Data Availability**

The data set used in the current study can be obtained from the corresponding author by an e-

mail request.



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

**Figure 1:** Map showing the study region. Estuaries of the rivers sampled in this study were indicated by solid black line.

**Figure 2:** Concentration (mg l$^{-1}$), export flux (Tg yr$^{-1}$) and yield (g m$^{-2}$ yr$^{-1}$) of dissolved inorganic carbon (DIC) in the Indian monsoonal estuaries.  Estuaries geographically located in the northeastern (NE), southeastern (SE), southwestern (SW) and northwestern (NW) regions of India were also shown.  Estuaries draining into the Bay of Bengal and the Arabian Sea were also provided

**Figure 3:** Spatial variability in stable carbon isotopes of dissolved inorganic carbon ($\delta^{13}C_{DIC}$, ‰) in the Indian monsoonal estuaries during discharge period.





**Figure 4:** (a) Positive correlation between dissolved inorganic carbon (DIC) concentration and
catchment area, and (b) negative correlation between DIC concentrations and annual mean
discharge ($km^3$) of the minor rivers.
**Figure 5:** Inverse correlation between mean dissolved inorganic carbon concentration in
estuaries (DIC, mg $l^{-1}$) and annual mean rainfall (mm) in catchments of the rivers in the NE, NW,
SE and SW regions of India.
**Figure 6:** Significant positive correlation between stable carbon isotopes of dissolved inorganic
carbon ($\delta^{13}C_{DIC}$, ‰) and salinity in the Indian monsoonal estuaries during the study period.
**Figure 7:** Significant positive correlation between stable carbon isotopes of dissolved inorganic
carbon ($\delta^{13}C_{DIC}$, ‰) and concentrations of DIC in the Indian monsoonal estuaries (filled
diamonds), SW estuaries (filled squares) and high saline estuaries (hollow triangles) during the
study period.
**Figure 8:** Relationship of dissolved inorganic carbon (DIC) yield (g $m^{-2}$ $yr^{-1}$) with that of (a)
rainfall (mm) and (b) soil organic carbon (kg $ha^{-1}$) in the catchment area of the NE, NW, SE and
SW rivers





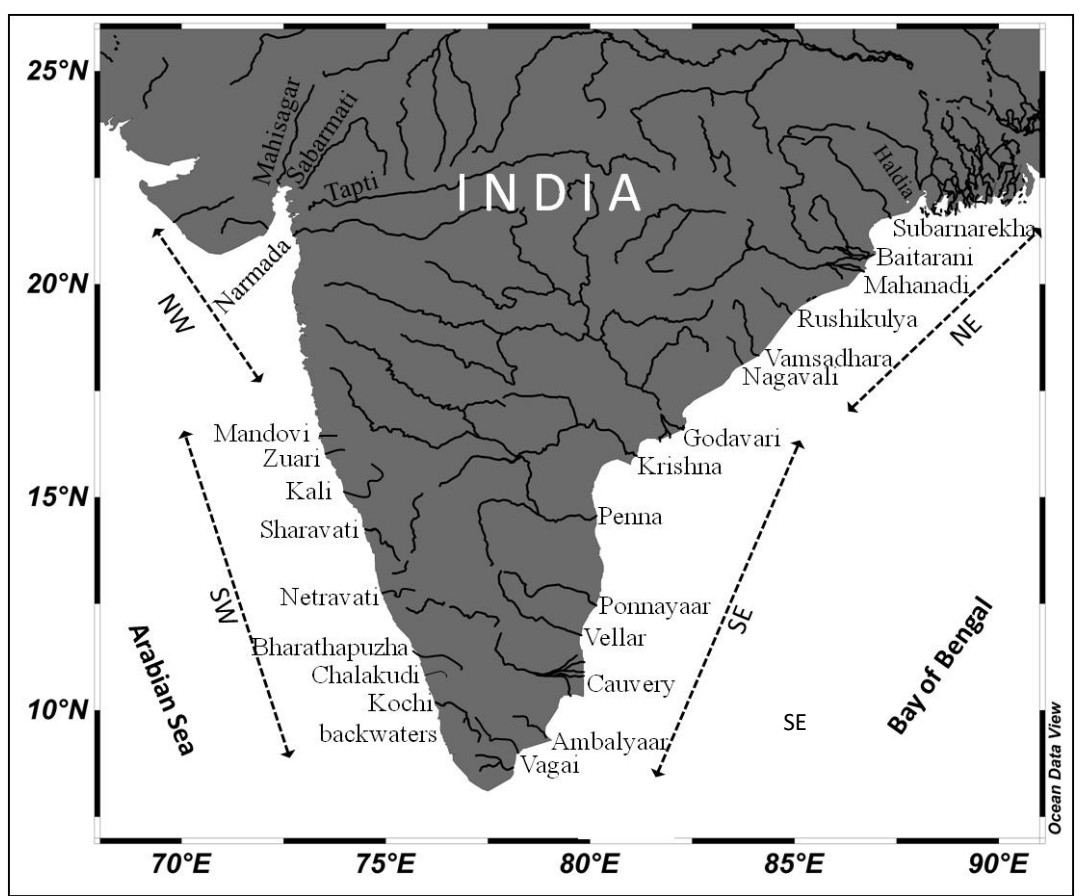


Fig.1:















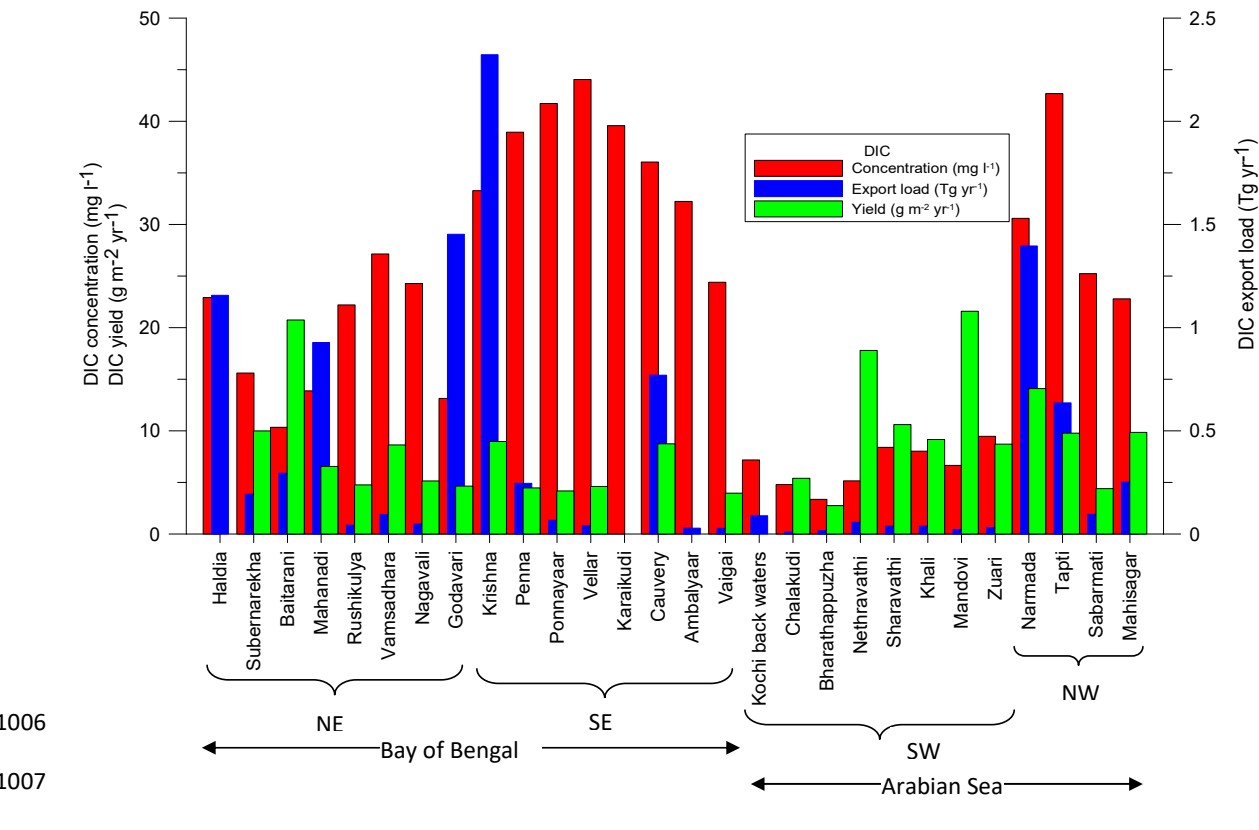



Fig. 2










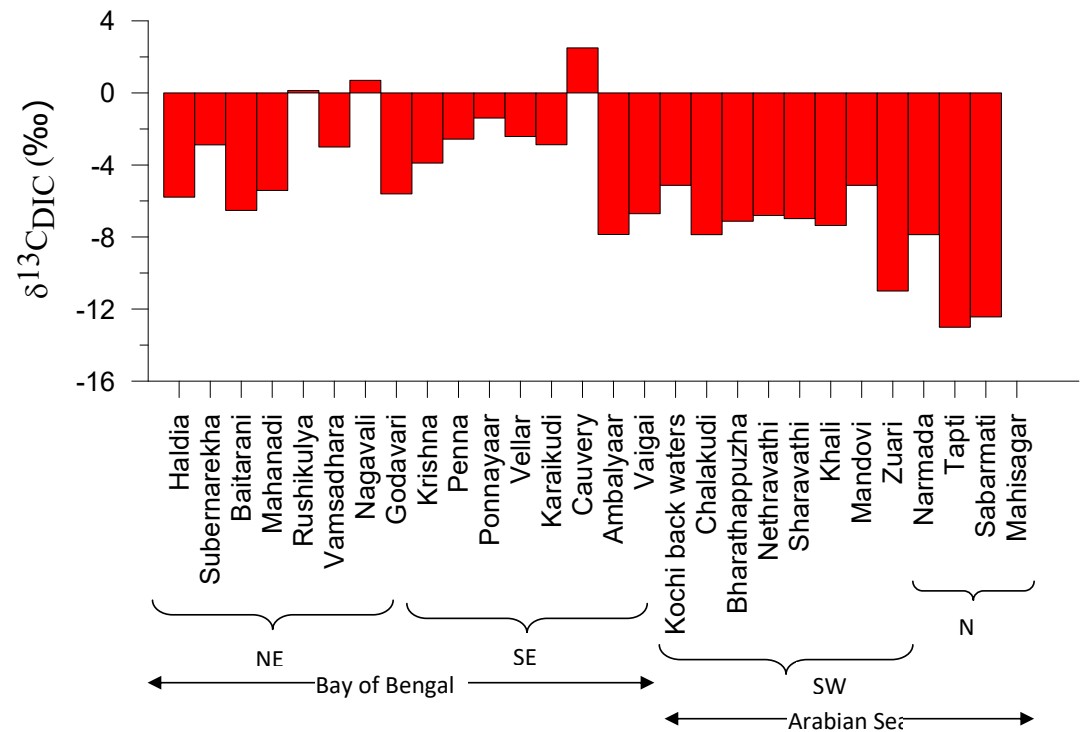




Fig. 3:














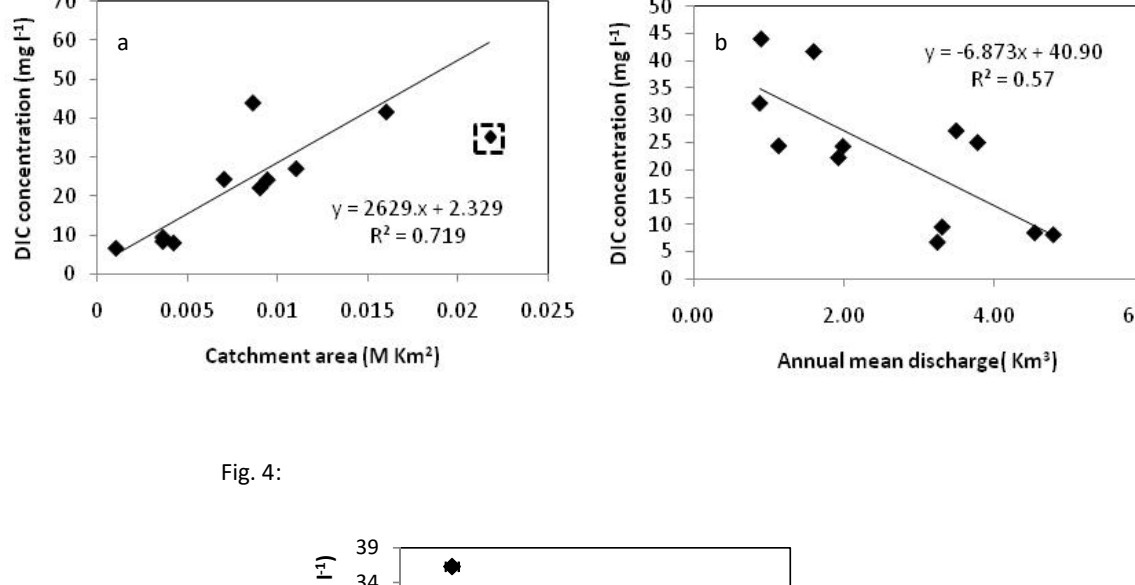


Fig. 4:


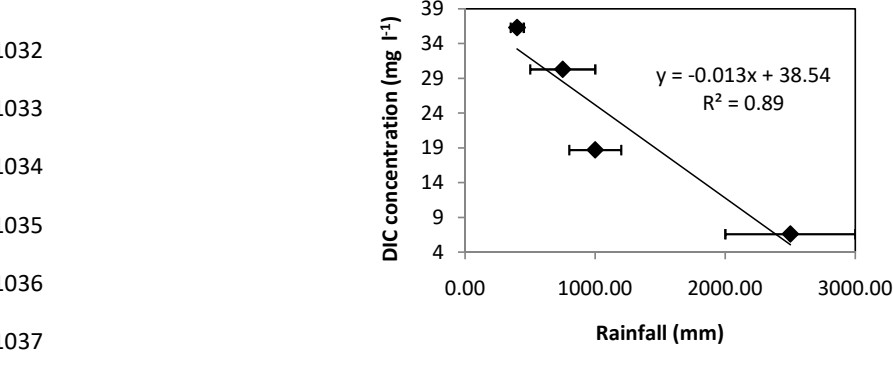

Fig. 5:

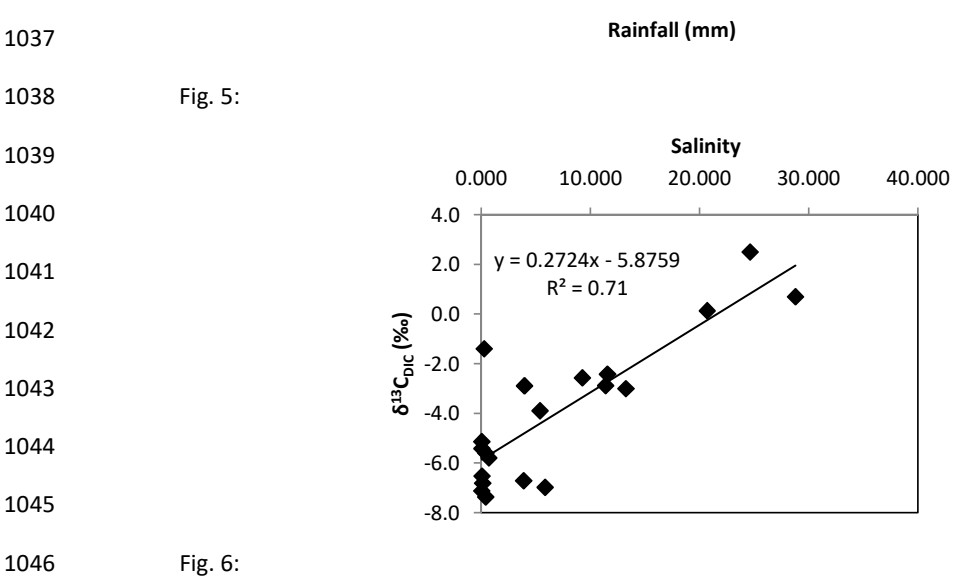

Fig. 6:












Fig. 7:

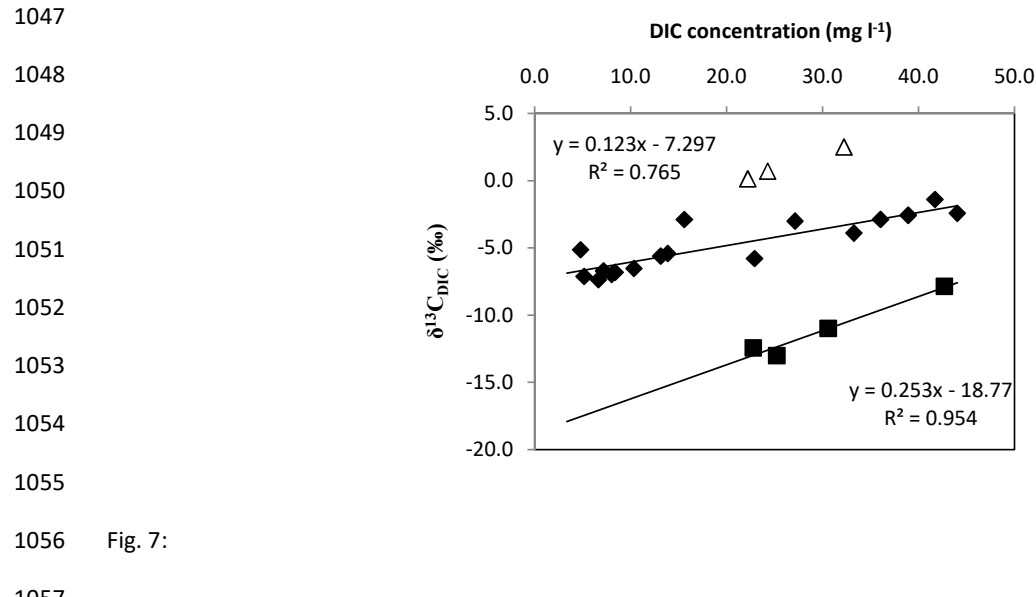




Fig. 8:

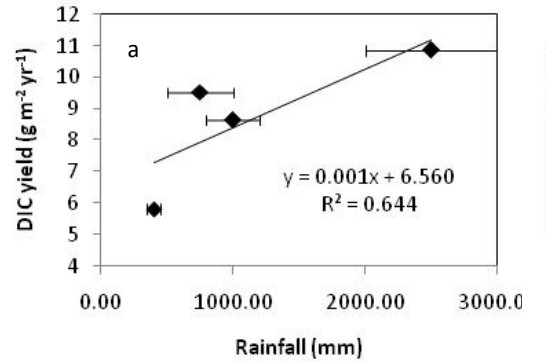

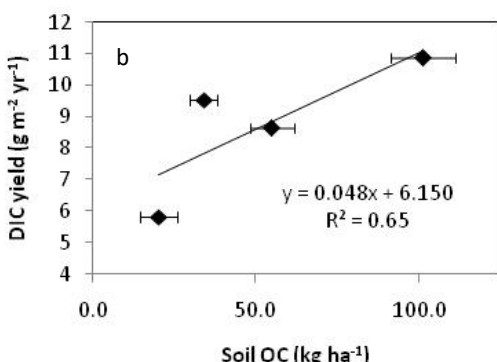

