# Peer review of "Export fluxes of dissolved inorganic carbon to the Northern Indian Ocean from the Indian monsoonal rivers"

_Biogeosciences, 2018_

## Referee Comment (RC1) · Anonymous Referee #1 · 9 Feb 2018

The paper by Krishna et al. provides a very large and almost complete data set on DIC concentrations in the estuaries of Indian rivers and the $\delta^{13}C_{DIC}$. The authors use the data to calculate DIC discharge to the Arabian Sea and Bay of Bengal and DIC yields from the catchments. It is a very important and valuable data set collected in 2011 and 2014 and should be published and made available also in the light of ongoing global change affecting the amount and sources of riverine DIC discharged to the ocean basins.

In the methods section the authors state that they have done multiple sampling in the estuaries. The standard deviation should be given in Figure 2.

In the methods the authors indicate that they have DO and chla data which they present only in a summarized form in Figures of the paper. These also need to be made available in an attachment.

The interpretation of the data is rather convoluted and therefore difficult to follow. It needs restructuring and would benefit from one or more tables. Furthermore, an introduction to the use of $\delta^{13}C_{DIC}$ as an indicator of DIC source is missing. Such information is given in lines 320-337. This could fit into the introduction.

Generally chapters 4.1 and 4.2 may be merged if the discussion is organized differently, may be as suggested below. Now some aspects of the $\delta^{13}C_{DIC}$ results are mentioned in 4.1 and 4.2 repeats some of the earlier arguments.

In order to better organize the discussion a Table would help showing the average rainfall in the four regions, the volume of discharge per m-2 and soil OC (lines 308 ff). To my mind the last paragraph of chapter 4.1. may rather be the starting point of the discussion.

In general I would suggest to follow a clear structure in the discussion (Chapter 4.1.), discussing consecutively (for example). (Most of these points are already mentioned in lines 237-245):
-   dilution effects and mixing effects with sea water (this point may well be discussed in the beginning to exclude certain samples from detailed source discussions using $\delta^{13}C_{DIC}$)
-   The impact of rock weathering: carbonate vs. silicates, which rock types dominate the catchment?
-   The impact of soil organic matter

In Schulte et al. I found a very good sketch of the contrasting impacts of these two mechanisms on $\delta^{13}C_{DIC}$ which may provide a helpful concept
-   Primary production and respiration in the river/catchment
-   possible anthropogenic impacts.
-   More points..?

The following collections of data could be given as Tables:
lines 225-233: Table of DIC concentrations selected rivers;
lines 390-396: Table of DIC discharge and
lines 415-418: Table of DIC yields of various regions/rivers.

Detailed comments:
Line 109ff: the studied rivers are perennial so that there is most probably some discharge during the non-monsoonal months and some of the river catchments may even receive winter rains so that these sentences have to be formulated a bit differently. The term "monsoonal rivers" is OK but the discharge during the other seasons may be stated as "small".
Lines 177/178: delete sentence
Lines 238-245: this is part of an introduction.
Lines 247 ff: the groups of rivers with high, intermediate and small discharges may be indicated.
Lines 255-260: this part is redundant, shorten.
Lines 320-337: introduction, see above. This part and the entire chapter have to be carefully checked. Atmospheric CO2 has a value of -7 to -8 ‰ but dissolved in water
Line 323: this is not clear: $CO_2$ has a $\delta^{13}C$ of -7 to -8 ‰ but when dissolved in water it is around 0 ‰ if the main anion is $HCO_3^-$.

Chapter 4.4 may be substantially shortened and I would suggest to shift the discussion on the source of DIC to the earlier chapter 4.1..

Lines 440ff is this contradicting the earlier discussion that groundwater is low in the SW and that this substantially is responsible for low concentrations?

Lines 450ff: the whole discussion on lithology could also be better in chapter 4.1.

May be a shorter discussion on the reason for the different yields (sediment/rock types and elevation; dams) would be sufficient.

The authors promise to send the data on request by E Mail. However, they should be made available in a data bank or as an attachment to the paper.

The abstract is quite good and may be retained even after changing the discussion. Likewise, changes of the summary would be also rather small after a revision.

Suggested Reference
Schulte, P., van Geldern, R., Freitag, H., Karim, A., Négrel, P., Petelet-Giraud, E., Probst, A., Probst, J.-L., Telmer, K., Veizer, J., and Barth, J. A. C.: Applications of stable water and carbon isotopes in watershed research: Weathering, carbon cycling, and water balances, Earth-Science Reviews, 109, 20-31, 2011.

---

## Referee Comment (RC2) · Anonymous Referee #2 · 7 Apr 2018

BG-2018-4 "Export fluxes of dissolved inorganic carbon to the Northern Indian Ocean from the Indian monsoonal rivers" by Moturi S. Krishna et al.

The manuscript by Moturi et al. presents an extensive database on the DIC fluxes to the Indian Ocean from the monsoonal rivers in the Indian subcontinent. It is clearly the outcome of very hard work, which resulted in this important dataset. As such, this work is valuable, and I can imagine that if published, this dataset would be used by modelers and other researchers. However, at its current form, the manuscript suffers from two essential drawbacks, which in my opinion, should be corrected before the manuscript can be published or even properly reviewed. My first and main concern is with the

quality of the presentation, namely the writing. The manuscript is heavily burdened by numerous grammatical mistakes, redundancies, and unintelligible sentences. Since English is not my native language, I am well aware of how hard it is to write in a foreign language, and therefore, I strongly urge the authors to have their manuscript edited by a native English speaker and/or by a professional editor. My second concern is with the somewhat superficial interpretation of the data. The authors relay heavily on correlations to investigate the relations between different characteristics of the rivers, but correlation do not necessarily imply cause/effect relations or, as the author argue in the discussion section. Therefore, the conclusions the authors draw are rather general, and do not go much beyond the data itself. Consequently, the manuscript has more resemblance to a report, and may be more appropriate to publication as such. I would recommend the authors to consider more carefully how this dataset can advance what we have already learned from previous works. At its present form, it is very hard to evaluate the scientific contribution of this work, and therefore, I recommend that this discussion paper be withdrawn, and perhaps submitted ab-initio after it has been thoroughly edited and revised.

General comments:

1) Grammar: The text is laden with grammatical errors. The first sentence in the abstract, for example, is flawed. So are lines 19-22, 65-68, 177-178 and many more. The usage of connectors ("However", "Though", "Despite" etc.) is wrong throughout the text. 2) Writing: Many sentences in the text are excessively long and incomprehensible (e.g. lines 30-35, lines 225-232). Reading sentences that contain more than 80 words, 11 values and more than 10 references (lines 225-232) is extremely demanding, and prevents the reader from understanding the messages that the authors try to convey. 3) Cumbersomeness and redundancies: Far too many results are incorporated in the text, instead of being presented as figures (e.g. lines 225-232, 247-250). This makes the manuscript cumbersome and turns the reading into a very demanding task. Some statements repeat themselves along the text (e.g. lines 176-177 and lines

216-217), making the text needlessly long. In some parts of the manuscript, there are no references to existing figures. Instead, the authors re-cite the values (see previous comment), whereas in others, the authors refer to relationships which should have been presented as figures (e.g. lines 263-264. See also detailed comment #22). 4) Units: The authors report most of their DIC data in mg l-1. This unit is somewhat archaic, and unclear. To what does the "mg" refer? Bicarbonate? Carbon? The more explicit concentration units of mol l-1 or mol kg-1 are much more common in the current literature. The authors themselves use mol kg-1 in the methods section. In the same section, they use percentage to describe the accuracy. This usage of multiple units for the same parameter is needlessly confusing. I recommend reporting all the results in mol l-1 or mol kg-1. 5) Error propagation and significant figures: In the methods sections, the authors report the analytical errors associated with their concentrations measurements. However, they do not propagate these errors to the DIC fluxes. In addition, the authors report too many significant figure compared to the error they report. 6) Figures and missing figures: Figures 1 and 2 are clear and informative. The rest of the figures are correlations, and could be presented in one or two panels. For some reason the authors did not include figures for some of the correlations they describe in the text. I cannot understand why.

Specific comments: (please note that I did not include all the grammatical errors in the text).

1) Line 10: change to "rivers are an/a important/significance source of..." 1) Line 19-22: revise this sentence. The usage of connectors is grammatically wrong. Use "enriched" instead of "caused the enrichment". Also, the "stable isotopic composition" cannot be "enriched". Use either "enriched in 13C" or "increase $\delta$13CDIC values" 2) Line 25: The sentence is grammatically wrong 3) Line 30: define "yield of DIC" 4) Lines 30-35: This sentence is too long and unintelligible 5) Line 56: "The Mississippi river" 6) Line 65: How do the fresh water discharge, and suspended sediment load relate to the fluvial carbon fluxes? 7) Line 67: The sentence is grammatically wrong 8) Line

71: change "estimating" to "estimations" 9) Lines 73-76: Most of the rivers mentioned in this paragraph are located between 30°S - 30°N. So why do the authors claim for "...paucity of data" (line 72) for this region? 10) Lines 76-81: The sentence is grammatically wrong 11) Line 82: The phrasing of this sentence is awkward, consider revising 12) Line 154: The units here are different from the units used in the text. Please be consistent. It is advised to use mol kg-1 throughout the text 13) Line 155: change "Scripts" to "Scripps" 14) Line 157: If the CRM from Andrew Dickson lab is used, 0.3 % equals approximately $\pm$ 6 $\mu$mol l-1. This is considerably larger than the precision the authors report in line 154. This error should be propagated along with other sources of error, to calculate the error on the flux estimations and DIC yield 15) Line 177-178: This sentence's phrasing is awkward, consider rephrasing 16) Line 179: If the error is in the second significant figure, it makes no sense to report 4 significant figure. Change 30.86$\pm$1.23 °C to 31$\pm$1 °C (and throughout the rest of the manuscript) 17) Line 205: remove the comma after "The estuaries" 18) Lines 216-217: This was already stated in lines 176-177. 19) Line 232: These values were already mentioned in line 224. 20) Line 236: I suggest that the mean values be added to figure 1 or to figure 2 21) Lines 247-250: There are way too many values in this sentence. 22) Lines 255-277: The authors describe 4 correlations here. None of them is shown in a figure, whereas other correlations are. Why did the authors chose not to show there correlations in figures? Since the readers cannot see the fit the authors used, there is no point in mentioning the (very poor) R2 values. 23) Line 328: The sentence is grammatically wrong 24) Line 501: Add the NIO number or remove this sentence

---

## Author Comment (AC1) · 28 Apr 2018

*Reviewer's comment*
The paper by Krishna et al. provides a very large and almost complete data set on DIC concentrations in the estuaries of Indian rivers and the δ13CDIC. The authors use the data to calculate DIC discharge to the Arabian Sea and Bay of Bengal and DIC yields from the catchments. It is a very important and valuable data set collected in 2011 and 2014 and should be published and made available also in the light of ongoing global change affecting the amount and sources of riverine DIC discharged to the ocean basins.

*Author's Response*
**Thank you very much.**

*Reviewer's comment*
In the methods section the authors state that they have done multiple sampling in the estuaries. The standard deviation should be given in Figure 2.

*Author's Response*
**Standard deviation will be given in Figure 2 as you suggested**

*Reviewer's comment*
In the methods the authors indicate that they have DO and chl-a data which they present only in a summarized form in Figures of the paper. These also need to be made available in an attachment.

*Author's Response*
**DO and Chl-a data will be made available in an attachment**

*Reviewer's comment*
The interpretation of the data is rather convoluted and therefore difficult to follow. It needs restructuring and would benefit from one or more tables.

*Author's Response*
**The interpretation (discussion) will be restructured as per your suggestion given in the next sections, and data will be provided in tables**

*Reviewer's comment*
Furthermore, an introduction to the use of $\delta^{13}C_{DIC}$ as an indicator of DIC source is missing. Such information is given in lines 320-337. This could fit into the introduction.

*Author's Response*
**Lines 320-327 will be shifted into the introduction section**

*Reviewer's comment*
Generally chapters 4.1 and 4.2 may be merged if the discussion is organized differently, may be as suggested below. Now some aspects of the $\delta^{13}C_{DIC}$ results are mentioned in 4.1 and 4.2 repeats some of the earlier arguments. In order to better organize the discussion a Table would help showing the average rainfall in the four regions, the volume of discharge per m-2 and soil OC (lines 308 ff). To my mind the last paragraph of chapter 4.1. may rather be the starting point of the discussion.

*Author's Response*

**The structure of the discussion will be modified by considering your suggestions. It will be started with the last paragraph of chapter 4.1, and the chapters 4.1 and 4.2 will be merged, to avoid repetitions, as you suggested.  Average rainfall, volume of discharge per m$^{-2}$ and soil OC data will be provided in a separate Table**

*Reviewer's comment*

In general I would suggest to follow a clear structure in the discussion (Chapter 4.1.), discussing consecutively (for example). (Most of these points are already mentioned in lines 237-245):

- dilution effects and mixing effects with sea water (this point may well be discussed in the   beginning to exclude certain samples from detailed source discussions using δ13CDIC)
- The impact of rock weathering: carbonate vs. silicates, which rock types dominate the catchment?
- The impact of soil organic matter

*Author's Response*

**The structure of the discussion will be re-organized in three subsections as you suggested. The first will be dedicated to the discussion on dilution and mixing effects, while the second will be dealt with impact of lithology (silicate versus carbonate rock weathering)  and the third will be focussed on soil organic matter.**

*Reviewer's comment*

In Schulte et al. I found a very good sketch of the contrasting impacts of these two mechanisms on $\delta^{13}C_{DIC}$ which may provide a helpful concept
- Primary production and respiration in the river/catchment
- possible anthropogenic impacts.
- More points..?

*Author's Response*

**Yes.  A very nice schematic it is. We will provide a schematic to explain the influence of different processes on increasing/decreasing the $\delta^{13}C_{DIC}$ in the Indian estuaries**

*Reviewer's comment*

The following collections of data could be given as Tables:
lines 225-233: Table of DIC concentrations selected rivers;
lines 390-396: Table of DIC discharge and
lines 415-418: Table of DIC yields of various regions/rivers.

*Author's Response*

**Data on DIC concentrations (L 225-233), total export (L. 390-396) and yield (L. 415-418) will be provided in the form of tables as you suggested.**

Detailed comments:

*Reviewer's comment*

Line 109ff: the studied rivers are perennial so that there is most probably some discharge during the non-monsoonal months and some of the river catchments may even receive winter

rains so that these sentences have to be formulated a bit differently. The term "monsoonal rivers" is OK but the discharge during the other seasons may be stated as "small".

*Author's Response*

**These rivers are not perennial and discharge from upstream rivers will not be there throughout the year. However, there is a small amount of discharge during the winter monsoon and it will be mentioned clearly in the revised submission.**

*Reviewer's comment*

Lines 177/178: delete sentence

*Author's Response*

**The sentence will be deleted as you suggested**

*Reviewer's comment*

Lines 238-245: this is part of an introduction.

*Author's Response*

**This part will be shifted to an introduction chapter**

*Reviewer's comment*

Lines 247 ff: the groups of rivers with high, intermediate and small discharges may be indicated.

*Author's Response*

**The groups of rivers based on their volume of discharge (high, intermediate and small) will be given**

*Reviewer's comment*

Lines 255-260: this part is redundant, shorten.

*Author's Response*

**This part will be shortened**

*Reviewer's comment*

Lines 320-337: introduction, see above. This part and the entire chapter have to be carefully checked. Atmospheric CO2 has a value of -7 to -8 ‰ but dissolved in water; Line 323: this is not clear: CO2 has a δ13C of -7 to -8 ‰ but when dissolved in water it is around 0 ‰ if the main anionis HCO3-

*Author's Response*

**Here, we meant to say that DIC originated by dissolution of atmospheric $CO_2$ (-7 to -8‰) is close to 0‰. We realized that it was presented wrongly. It will be corrected in the revision.**

*Reviewer's comment*

Chapter 4.4 may be substantially shortened and I would suggest to shift the discussion on the source of DIC to the earlier chapter 4.1.

*Author's Response*

**As we mentioned above, the discussion related to sources of DIC, for example, lithology, in this chapter will be shifted to chapter 4.1. The discussion on yield of DIC will be shortened and will be given in a paragraph.**

*Reviewer's comment*

Lines 440ff is this contradicting the earlier discussion that groundwater is low in the SW and that this substantially is responsible for low concentrations?

*Author's Response*

**Here, we mean to say that higher yield of DIC from SW estuaries may not be due to large contribution from ground water as their concentration were found to low than the other Indian estuaries of NW, SW and NE region. However, to obtain the clarity these two sections will be modified.**

*Reviewer's comment*

Lines 450ff: the whole discussion on lithology could also be better in chapter 4.1.
May be a shorter discussion on the reason for the different yields (sediment/rock types and elevation; dams) would be sufficient.

*Author's Response*

**The discussion on lithology will be shifted to chapter 4.1 as you suggested. Also, the main reasons for different yields (rock type, dams and soils organic matter) will be provided in a short paragraph.**

*Reviewer's comment*

The authors promise to send the data on request by E Mail. However, they should be made available in a data bank or as an attachment to the paper.

*Author's Response*

**Data is accessible through the website of our Institute's data centre (http://www.nio.org/iodc)**

*Reviewer's comment*

The abstract is quite good and may be retained even after changing the discussion. Likewise, changes of the summary would be also rather small after a revision.

*Author's Response*

**Thank you. Summary will be refined.**

---

## Author Comment (AC2) · 28 Apr 2018

*Reviewer's comment*
The manuscript by Moturi et al. presents an extensive database on the DIC fluxes to the Indian Ocean from the monsoonal rivers in the Indian subcontinent. It is clearly the outcome of very hard work, which resulted in this important dataset. As such, this work is valuable, and I can imagine that if published, this dataset would be used by modelers and other researchers.

*Author's Response*
**Thank you very much.**

*Reviewer's comment*
However, at its current form, the manuscript suffers from two essential drawbacks, which in my opinion, should be corrected before the manuscript can be published or even properly reviewed.

*Author's Response*
**The manuscript will be modified/corrected to overcome these two drawbacks you mentioned**

*Reviewer's comment*
My first and main concern is with the quality of the presentation, namely the writing. The manuscript is heavily burdened by numerous grammatical mistakes, redundancies, and unintelligible sentences. Since English is not my native language, I am well aware of how hard it is to write in a foreign language, and therefore, I strongly urge the authors to have their manuscript edited by a native English speaker and/or by a professional editor.

*Author's Response*
**Since English is not our native language, we have a limited control on English language and therefore there are mistakes in grammar, phrasing, syntax and connecting words in the manuscript. We would like to use the services of an expert in English language editing and therefore we will submit the revised manuscript with a better English language**

*Reviewer's comment*
My second concern is with the somewhat superficial interpretation of the data. The authors relay heavily on correlations to investigate the relations between different characteristics of the rivers, but correlation do not necessarily imply cause/effect relations or, as the author argue in the discussion section. Therefore, the conclusions the authors draw are rather general, and do not go much beyond the data itself. Consequently, the manuscript has more resemblance to a report, and may be more appropriate to publication as such. I would recommend the authors to consider more carefully how this dataset can advance what we have already learned from previous works. At its present form, it is very hard to evaluate the scientific contribution of this work, and therefore, I recommend that this discussion paper be withdrawn, and perhaps submitted ab-initio after it has been thoroughly edited and revised.

*Author's Response*
**Since one of the objectives of this work is to understand the influence of river/catchment characteristics on the export and yield of DIC, we have used correlations between various parameters to explain the role or impact of different characteristics of rivers on**

**the export and yield of DIC from the Indian monsoonal rivers to the northern Indian Ocean. Though the correlation do not necessarily imply cause/effect relations, as you said, we performed the correlations to check the statistical significance of river characteristics on DIC export to the northern Indian Ocean. Here, we made an attempt to identify the statistically significant parameters controlling the DIC export by the Indian monsoonal rivers which will be subsequently used by the modellers, as you mentioned earlier, and it is not in the scope of this paper.**

*Reviewer's comment*
The text is laden with grammatical errors. The first sentence in the abstract, for example, is flawed. So are lines 19-22, 65-68, 177-178 and many more. The usage of connectors ("However", "Though", "Despite" etc.) is wrong throughout the text.

*Author's Response*
**Mistakes associated with English language such as grammar, phrasing, syntax and connecting words will be rectified in the revised submission as we would like to use the services of an expert in English language.**

*Reviewer's comment*
Many sentences in the text are excessively long and incomprehensible (e.g. lines 30-35, lines 225-232). Reading sentences that contain more than 80 words, 11 values and more than 10 references (lines 225-232) is extremely demanding, and prevents the reader from understanding the messages that the authors try to convey.

*Author's Response*
**Long and incomprehensible sentences will be modified to obtain the clarity in conveying the message to readers.**

*Reviewer's comment*
Cumbersomeness and redundancies: Far too many results are incorporated in the text, instead of being presented as figures (e.g. lines 225-232, 247-250). This makes the manuscript cumbersome and turns the reading into a very demanding task.

*Author's Response*
**Results mentioned in the text will be deleted to maintain the focus of the manuscript. These results will be provided in a table.**

*Reviewer's comment*
Some statements repeat themselves along the text (e.g. lines 176-177 and lines 216-217), making the text needlessly long.

*Author's Response*
**Repeated statements will be deleted throughout the manuscript**

*Reviewer's comment*
In some parts of the manuscript, there are no references to existing figures. Instead, the authors re-cite the values (see previous comment), whereas in others, the authors refer to relationships which should have been presented as figures (e.g. lines 263-264. See also detailed comment #22).

*Author's Response*
**All figures will be cited in text. All relationships mentioned in the text will be given as figures in one or two panels.**

*Reviewer's comment*
Units: The authors report most of their DIC data in mg l-1. This unit is somewhat archaic, and unclear. To what does the "mg" refer? Bicarbonate? Carbon? The more explicit concentration units of mol l-1 or mol kg-1 are much more common in the current literature. The authors themselves use mol kg-1 in the methods section. In the same section, they use percentage to describe the accuracy. This usage of multiple units for the same parameter is needlessly confusing. I recommend reporting all the results in mol l-1 or mol kg-1.

*Author's Response*
**Unit mol kg$^{-1}$ will be used throughout the text as you suggested**

*Reviewer's comment*
Error propagation and significant figures: In the methods sections, the authors report the analytical errors associated with their concentrations measurements. However, they do not propagate these errors to the DIC fluxes. In addition, the authors report too many significant figure compared to the error they report.

*Author's Response*
**The errors associated with flux and yield estimations will be given and provided in Figures 1 and 2 also in the revised submission. Significant figure will also be followed.**

*Reviewer's comment*
Figures and missing figures: Figures 1 and 2 are clear and informative. The rest of the figures are correlations, and could be presented in one or two panels. For some reason the authors did not include figures for some of the correlations they describe in the text. I cannot understand why.

*Author's Response*
**All correlations will be presented in the form of figures in one or two panels as you suggested**

*Reviewer's comment*
Line 10: change to "rivers are an/a important/significance source of: : :"

*Author's Response*
**It will be changed to 'rivers are an important source of' ...**

*Reviewer's comment*
Line 19-22: revise this sentence. The usage of connectors is grammatically wrong. Use "enriched" instead of "caused the enrichment". Also, the "stable isotopic composition" cannot be "enriched". Use either "enriched in 13C" or "increase _13CDIC values"

*Author's Response*
**The sentence will be revised and 'enriched in $^{13}$C' will be used as you suggested**

*Reviewer's comment*
Line 25: The sentence is grammatically wrong

*Author's Response*
**The sentence will be corrected**

*Reviewer's comment*
Line 30: define "yield of DIC"

*Author's Response*
**DIC will be defined**

*Reviewer's comment*
Lines30-35: This sentence is too long and unintelligible

*Author's Response*
**The sentence will be shortened and may be presented in two sentences**

*Reviewer's comment*
Line 56: "The Mississippi river"

*Author's Response*
**It will be corrected**

*Reviewer's comment*
Line 65: How do the fresh water discharge, and suspended sediment load relate to the fluvial carbon fluxes?

*Author's Response*
**Sediment load will be deleted as it will not directly influence the fluvial carbon fluxes. However, freshwater discharge significantly influences the fluvial carbon fluxes to estuaries and coastal region as it scours terrestrial carbon from carbonate rocks and soils. Nevertheless, this sentence will be modified in the revision to convey the message more clearly**

*Reviewer's comment*
Line 67: The sentence is grammatically wrong

*Author's Response*
**The sentence will be corrected**

*Reviewer's comment*
Line 71: change "estimating" to "estimations"

*Author's Response*
**It will be changed to 'estimations'**

*Reviewer's comment*
Lines 73-76: Most of the rivers mentioned in this paragraph are located between $30^{o}$S - $30^{o}$N. So why do the authors claim for ": : :paucity of data" (line 72) for this region?

*Author's Response*
**Here, we meant to say that many of the medium rivers from this region were not included in the global DIC estimations due to the paucity of data. The rivers mentioned in lines 73-76 (Mississippi, Congo, Changjiang and Pearl) are the large rivers in the world. However, to obtain the clarity, these sentences will be modified.**

*Reviewer's comment*
Lines 76-81: The sentence is grammatically wrong

*Author's Response*
**The sentence will be corrected**

*Reviewer's comment*
Line 82: The phrasing of this sentence is awkward, consider revising

*Author's Response*
**The phrasing of this sentence will be modified (rephrased)**

*Reviewer's comment*
Line 154: The units here are different from the units used in the text. Please be consistent. It is advised to use mol kg-1 throughout the text

*Author's Response*
**Unit 'mol kg$^{-1}$' will be used throughout the text as you suggested**

*Reviewer's comment*
Line 155: change "Scripts" to "Scripps"

*Author's Response*
**Sorry for the mistake. It will be corrected**

*Reviewer's comment*
Line 157: If the CRM from Andrew Dickson lab is used, 0.3 % equals approximately ±6 µmol l-1. This is considerably larger than the precision the authors report in line 154. This error should be propagated along with other sources of error, to calculate the error on the flux estimations

*Author's Response*
**Here, we mentioned the precision but not accuracy. As mentioned earlier, errors associated with determination of DIC concentrations and the other errors will be propagated to DIC export flux and yield calculations. These errors will be shown in figures (1 and 2) also**.

*Reviewer's comment*
Line 177-178: This sentence's phrasing is awkward, consider rephrasing

*Author's Response*
**The sentence will be rephrased as you suggested**

*Reviewer's comment*
Line 179: If the error is in the second significant figure, it makes no sense to report 4 significant figure. Change 30.86±1.23 °C to 31±1 °C (and throughout the rest of the manuscript)

*Author's Response*
**It will be corrected during revision**

*Reviewer's comment*
Line 205: remove the comma after "The estuaries"

*Author's Response*
**Comma will be removed after "The estuaries"**

*Reviewer's comment*
Lines 216-217: This was already stated in lines 176-177.

*Author's Response*
**The sentence will be deleted to avoid repetition**

*Reviewer's comment*
Line 232: These values were already mentioned in line 224.

*Author's Response*
**These values will be deleted to avoid repetition**

*Reviewer's comment*
Line 236: I suggest that the mean values be added to figure 1 or to figure 2

*Author's Response*
**Mean values will be added in figures 1 and 2 as you suggested**

*Reviewer's comment*
Lines 247-250: There are way too many values in this sentence.

*Author's Response*
**All these values will be deleted from the text and will be given in a table**

*Reviewer's comment*
Lines 255-277: The authors describe 4 correlations here. None of them is shown in a figure, whereas other correlations are. Why did the authors chose not to show there correlations in figures?

*Author's Response*
**All correlations will be provided as figures**

*Reviewer's comment*
Since the readers cannot see the fit the authors used, there is no point in mentioningthe (very poor) $R^2$ values.

*Author's Response*

**Fit will be provided in the form of figures in two panels as you suggested above**

*Reviewer's comment*

Line 328: The sentence is grammatically wrong

*Author's Response*

**The sentence will be corrected**

*Reviewer's comment*

Line 501: Add the NIO number or remove this sentence

*Author's Response*

**Contribution number will be added only after the manuscript has been accepted for publication (during galley proof correction).**

---

## Referee Comment (RC3) · G. V. M. Gupta (Referee) · 9 May 2018

bg-2018-4 "Export fluxes of dissolved inorganic carbon to the northern Indian Ocean from the Indian monsoonal rivers" by Krishna et al.

This paper presents a hard work from the extensive field coverage of 27 Indian monsoonal estuaries twice during the discharge period of two different years. In the growing concern of climate change when many of the biophysical and biogeochemical models are suffering from the lack of data sets from the tropical rivers, I am sure this paper once published will significantly fill that gap and heavily used by many researchers. However, the manuscript requires to provide clarity and corrections on certain issues before it is published.

The DIC concentrations and fluxes are influenced by the rainfall variability among the four regions, the discussion will be benefited if it starts with this information. From Figure 1, it is apparent that many of the east flowing rivers, especially in the central and southern regions, are sourced from the western catchments but none in the vice versa direction. This is important and highlighted because high rainfall SW regions have less discharge and DIC fluxes but much of this rainfall might be sourcing the less rain fed SE rivers and contribute to high DIC fluxes.

I strongly suggest the authors to include a Table of all rivers sampled (grouped into four regions) with details of their size-class (large and medium), catchment size, length of the river, soil organic carbon, discharge rate, mean DIC concentration, export flux, yield, etc. for better utilizing the hard work of this study by scientific community.

Many of the statements are repeated throughout the manuscript which makes it length, for example, parts of section 4.2 and 4.4 carry some common information. Restructuring of discussion by appropriately merging relevant subsections will improve the focus and clarity. Number of figures can also be minimized, for example, merge figs.4 & 5 and 6 & 7.

The manuscript requires thorough editing for English grammar for better reading.

Specific Comments:

Line 43: delete 'about'.
Line 54: it is an obvious statement, delete.
Lines 55-57: how much increase? Specify 'Mississippi river'.
Lines 76-81: include carbon studies from Gupta et al. (2008) in the Chilka lake, a brackish water estuarine system. Also, include Bhavya et al. (2018) for Cochin estuary.
Lines 81-82: Carbon export fluxes from the Chilka lake (Gupta et al., 2008) and Cochin estuary (Gupta et al., 2009) on east and west coast of India respectively were earlier reported.
Lines 95-102 & 120-124: Too big sentences.
Lines 132-134: Year 2011 was a normal monsoon year but 2014 was an El-Nino year. Please comment or speculate the variability in light of having used discharge data of

earlier years from the published literature. Authors may refer to Indian Annual Rainfall Statistics reports available online at www.imd.gov.in.

Lines 134-137: These are contradicting the statements made at lines 130-131.

Line 139: replace was with 'were'.

Line 174: specify the source of catchment area.

Lines 185-186: give mean±SD values.

Lines 207-208: delete 'by the Indian monsoonal rivers'.

Lines 216-17: repeated statement

Lines 250-254: Provide full details in a Table for better usage of this work by many researchers.

Line 256: Include Gupta et al., 2008 for Chilka lake. Bhavya et al. 2016 covers only dry season (postmonsoon), replace it with Bhavya et al. 2018 for all seasons.

Lines 260-262: Rather relationship with TOC (DOC+POC) is better.

Lines 282-286: It seems this ground water regional variation is following the variability of DIC in the regional estuaries. Does this mean the cause factors for DIC variation are also applicable for its variation in the ground water? Please make a statement on this.

Line 285: provide units for all the values.

Lines 286-289: Grammatically sentence not correct.

Lines 304-307: Please comment, if not speculate, on whether these soil characteristics are limited only to surface or extended to the vertical strata as well, which can give an insight into whether the source of low DIC in these surface and ground waters are same or different.

Lines 310-312: Weathering rates may be high due to highest precipitation but DIC flux from the weathering of lateritic soils to the SW estuaries (refer lines 304-307) could have been far lower than other regions.

Lines 316-318: better integrate these with statements made at lines 365-471 and attribute to intense precipitation, presence of less weathering lateritic soils and soil organic carbon.

Lines 325-330: both the statements correspond to the weathering but the contribution of $\delta13C_{DIC}$ values were differently reported. Pls check.

Lines 352-355: Repetition of statements at lines 326-330 but with clarity here. Avoid repetition.

Lines 395-396: Are these discharge per day or year?

Line 401: Relatively higher export fluxes…….. compared to what?

Line 405: When combined…..with DIC export flux?

Lines 424-425: SW region is having highest rainfall but lowest discharge rate from smallest catchement area. If so, large amount of rainfall might be happening over the non-catchment area. What would be the fate of this? Please discuss on the possibility of its seeping into the ground water and its contribution of DIC flux to the SW coast of India, its relativity with respect to surface flux?

Lines 432-433: include -ve sign for the r2 values for having the negative relationships.

Lines 441-442: Reference to the comment for lines 424-425. Does low DIC concentration in the ground water of SW region is also due to high dilution rate and possible lateritic soil strata? Please comment on what would be the ground water discharge rate and its associated DIC export flux to the SW coastal AS compared to the other regions.

Line 469: soil organic carbon content….what is the source for this data?

Suggested Literature:

1. Bhavya, P.S., Sanjeev Kumar, Gupta, G.V.M., Sudharma, K.V., Sudheesh, V. (2018). Spatio-temporal variation in $\delta^{13}C_{DIC}$ of a tropical eutrophic estuary (Cochin estuary, India) and adjacent Arabian Sea. *Continental Shelf Research*, 153, 75-85, doi: 10.1016/j.csr.2017.12.006.
2. Gupta, G.V.M., Sarma, V.V.S.S., Robin, R.S., Raman, A.V., Jai Kumar, M., Rakesh, M. and Subramanian, B.R (2008). Influence of net ecosystem metabolism in transferring riverine organic carbon to atmospheric $CO_2$ in a tropical coastal lagoon (Chilka Lake, India). *Biogeochemistry,* 87: 265-285, doi:10.1007/s10533-008-9183-x.

---

## Author Comment (AC3) · 31 May 2018

*Reviewers comment*
This paper presents a hard work from the extensive field coverage of 27 Indian monsoonal estuaries twice during the discharge period of two different years. In the growing concern of climate change when many of the biophysical and biogeochemical models are suffering from the lack of data sets from the tropical rivers, I am sure this paper once published will significantly fill that gap and heavily used by many researchers.

*Author's response*
**Thank you very much**

*Reviewers comment*
However, the manuscript requires to provide clarity and corrections on certain issues before it is published.

*Author's response*
**The manuscript will be revised to improve the clarity as per your suggestions/comments**

*Reviewers comment*
The DIC concentrations and fluxes are influenced by the rainfall variability among the four regions, the discussion will be benefited if it starts with this information.

*Author's response*
**Yes. Concentration and fluxes of riverine DIC are strongly influenced by the variability in rainfall over catchment of the river (region). This information will be provided in the initial part of the discussion during restructuring the discussion part of the manuscript.**

*Reviewers comment*
From Figure 1, it is apparent that many of the east flowing rivers, especially in the central and southern regions, are sourced from the western catchments but none in the vice versa direction. This is important and highlighted because high rainfall SW regions have less discharge and DIC fluxes but much of this rainfall might be sourcing the less rain fed SE rivers and contribute to high DIC fluxes. I strongly suggest the authors to include a Table of all rivers sampled (grouped into four regions) with details of their size-class (large and medium), catchment size, length of the river, soil organic carbon, discharge rate, mean DIC concentration, export flux, yield, etc. for better utilizing the hard work of this study by scientific community.

*Author's response*
**A table containing the information of all rivers including their size-class (large and medium), catchment area, length of the river, soil organic carbon, discharge rate, mean DIC concentration, export flux, yield, etc will be provided as suggested by you (and also the Reviewer 1)**

*Reviewers comment*
Many of the statements are repeated throughout the manuscript which makes it length, for example, parts of section 4.2 and 4.4 carry some common information. Restructuring of discussion by appropriately merging relevant subsections will improve the focus and clarity.

*Author's response*
**The manuscript will be restructured to avoid repetitions and to increase the focus and clarity of the manuscript. Reviewer 1 also suggested to re-structure the manuscript to avoid some repetitions.**

*Reviewers comment*
Number of figures can also be minimized, for example, merge figs.4 & 5 and 6 &7.

*Author's response*
**Figures 4&5 and 6&7 will be merged as you suggested**

*Reviewers comment*
The manuscript requires thorough editing for English grammar for better reading.

*Author's response*
**The revised manuscript will be proof read by the English language expert**

**Specific Comments:**

*Reviewers comment*
Line 43: delete 'about'.

*Author's response*
 **'about' will be deleted.**

*Reviewers comment*
Line 54: it is an obvious statement, delete.

*Author's response*
**The sentence will be deleted**

*Reviewers comment*
Lines 55-57: how much increase? Specify 'Mississippi river'.

*Author's response*
**As per *Ren et al.,* (2015) the total increase in DIC export throughout the 21st century from the Mississippi River to Gulf of Mexico would be over 90% due to the combined effect of climate-related changes along with rising atmospheric $CO_2$ . This will be mentioned in the revised manuscript to obtain clarity.**

*Reviewers comment*
Lines 76-81: include carbon studies from Gupta et al. (2008) in the Chilka lake, a brackish water estuarine system. Also, include Bhavya et al. (2018) for Cochin estuary.

*Author's response*
**Chilka lake (*Gupta et al.,* 2008) and Cochin estuary (*Bhavya et al.,* 2018) studies will be included**

*Reviewers comment*
Lines 81-82: Carbon export fluxes from the Chilka lake (Gupta et al., 2008) and Cochin estuary (Gupta et al., 2009) on east and west coast of India respectively were earlier reported.

*Author's response*
**A sentence "Carbon export fluxes from the Chilka lake (Gupta et al., 2008) and Cochin estuary (Gupta et al., 2009) on east and west coast of India respectively were earlier reported" will be added here (L 81-82) as you suggested.**

*Reviewers comment*
Lines 95-102 & 120-124: Too big sentences.

*Author's response*
**These sentences will be modified.**

*Reviewers comment*
Lines 132-134: Year 2011 was a normal monsoon year but 2014 was an El-Nino year.

*Author's response*
**The mean values of normal monsoon and weak monsoon (El Nino) provides better mean concentrations rather than the mean of two normal monsoon years (expected to be higher side than long term mean) or two weak monsoon years (expected to be lower than long term mean). Therefore, field sampling in this study was conducted one during normal monsoon year and the other during weak monsoon year**

*Reviewers comment*
Please comment or speculate the variability in light of having used discharge data of earlier years from the published literature. Authors may refer to Indian Annual Rainfall Statistics reports available online at www.imd.gov.in.

*Author's response*
**We will consider the Annual Rainfall Statistics report from IMD, New Delhi for discharge data.**

*Reviewers comment*
Lines 134-137: These are contradicting the statements made at lines 130-131.

*Author's response*
**Lines 130-131 mean to say that it is from starting point (origin) to ending point (estuary) of the river, i.e. entire length of the river; whereas Line 134-137 means that it is the length of the estuary (upper and lower estuaries) but not the entire length of the river. However, these sentences will be modified to obtain clarity.**

*Reviewers comment*
Line 139: replace was with 'were'.

*Author's response*
**Sorry for the mistake. 'was' will be replaced with 'were'**

*Reviewers comment*
Line 174: specify the source of catchment area.

*Author's response*
**Source of the catchment area will be provided.**

*Reviewers comment*
Lines 185-186: give mean±SD values.

*Author's response*
**Mean±SD values will be provided**

*Reviewers comment*
Lines 207-208: delete 'by the Indian monsoonal rivers'.

*Author's response*
**'by the Indian monsoonal rivers' will be deleted as you suggested**

*Reviewers comment*
Lines 216-17: repeated statement

*Author's response*
**The repeated statement will be deleted**

*Reviewers comment*
Lines 250-254: Provide full details in a Table for better usage of this work by many researchers.

*Author's response*
**A table will be provided with complete details as mentioned above**

*Reviewers comment*
Line 256: Include Gupta et al., 2008 for Chilka lake. Bhavya et al. 2016 covers only dry season (postmonsoon), replace it with Bhavya et al. 2018 for all seasons.

*Author's response*
**Gupta et al., 2008 will be included for Chilka lake. *Bhavya et al.*, 2016 will be replaced with *Bhavya et al.*, 2018.**

*Reviewers comment*
Lines 260-262: Rather relationship with TOC (DOC+POC) is better.

*Author's response*
**The relationship with TOC will be examined and will be provided as you suggested.**

*Reviewers comment*
Lines 282-286: It seems this ground water regional variation is following the variability of DIC in the regional estuaries. Does this mean the cause factors for DIC variation are also applicable for its variation in the ground water? Please make a statement on this.

*Author's response*

**Since ground waters are one of the important sources of DIC in estuaries, it is possible that ground water DIC concentrations will have significant impact on DIC concentrations in estuaries. However, due to the influence of other factors such as hydrology, lithology and environmental characteristics of the catchment on DIC concentrations in estuaries, it is very difficult to make a statement that only ground water is the cause factor for variability of DIC concentrations in estuaries.**

*Reviewers comment*

Line 285: provide units for all the values.

*Author's response*

**Units will be provided for all the values as you suggested**

*Reviewers comment*

Lines 286-289: Grammatically sentence not correct.

*Author's response*

**The sentence will be corrected**

*Reviewers comment*

Lines 304-307: Please comment, if not speculate, on whether these soil characteristics are limited only to surface or extended to the vertical strata as well, which can give an insight into whether the source of low DIC in these surface and ground waters are same or different.

*Author's response*

**We considered the characteristics (lithology) of only surface rocks/soils in the catchment area of the river/region from soil maps of India available in literature. We have not discussed the vertical strata of the rocks/soils. It is possible that vertical strata of the rocks could have influenced the low DIC concentrations in ground waters of the SW region. However, we have not focussed on the reasons for spatial variability in ground water DIC concentration as it is not the scope of this study.**

*Reviewers comment*

Lines 310-312: Weathering rates may be high due to highest precipitation but DIC flux from the weathering of lateritic soils to the SW estuaries (refer lines 304-307) could have been far lower than other regions.

*Author's response*

**DIC concentrations and export flux are far lower in the SW region than the other regions and it could be due to, at least partly, the dominance of lateritic soils in the catchment of SW rivers. However, the dense rainfall over the SW region increases the scouring of DIC from soils and therefore causes elevated yield of DIC (DIC export per unit area of catchment) from SW rivers.**

*Reviewers comment*

Lines 316-318: better integrate these with statements made at lines 365-471 and attribute to intense precipitation, presence of less weathering lateritic soils and soil organic carbon.

*Author's response*

**This will be corrected and statements will be integrated during re-structuring the discussion part of the manuscript.**

*Reviewers comment*

Lines 325-330: both the statements correspond to the weathering but the contribution of d13CDIC values were differently reported. Pls check.

*Author's response*

**Though, both the statements correspond to weathering of rocks, the resulted $\delta^{13}C$ of DIC is different. This is because the $\delta^{13}C$ of carbonic acid formed by dissolution of soil $CO_2$ is different from that of the $\delta^{13}C$ of carbonic acid formed by dissolution of atmospheric $CO_2$. However, this will be mentioned in the text for clarity during revision.**

*Reviewers comment*

Lines 352-355: Repetition of statements at lines 326-330 but with clarity here. Avoid repetition.

*Author's response*

**Repeated sentences will be deleted**

*Reviewers comment*

Lines 395-396: Are these discharge per day or year?

*Author's response*

**These discharges are per year. This will be mentioned in the text for clarity during revision.**

*Reviewers comment*

Line 401: Relatively higher export fluxes…….. compared to what?

*Author's response*

**It is compared to global rivers. The sentence will be modified to obtain clarity.**

*Reviewers comment*

Line 405: When combined…..with DIC export flux?

*Author's response*

**'with DIC export flux' will be added after 'When combined' at Line 405 as you suggested.**

*Reviewers comment*

Lines 424-425: SW region is having highest rainfall but lowest discharge rate from smallest catchment area. If so, large amount of rainfall might be happening over the non-catchment area. What would be the fate of this? Please discuss on the possibility of its seeping into the ground water and its contribution of DIC flux to the SW coast of India, its relativity with respect to surface flux?

*Author's response*

**The density of rainfall is high in the SW than the other regions of India. However, lower river discharge from the SW rivers is mainly due to small catchment area of the SW rivers than the other peninsular rivers. Though we have discussed the influence of ground water DIC concentrations on the export flux and yield of DIC from the Indian peninsular rivers, we could not quantify the ground water DIC flux to the coastal waters (Arabian Sea and Bay of Bengal) and their contribution to surface DIC export flux as we have not determined the ground water exchange rates, which is not within the scope of the present study.**

*Reviewers comment*
Lines 432-433: include -ve sign for the r2 values for having the negative relationships.

*Author's response*
**Negative sign '-' will be given for the $r^2$ value, if the relationship is inverse.**

*Reviewers comment*
Lines 441-442: Reference to the comment for lines 424-425. Does low DIC concentration in the ground water of SW region is also due to high dilution rate and possible lateritic soil strata? Please comment on what would be the ground water discharge rate and its associated DIC export flux to the SW coastal AS compared to the other regions.

*Author's response*
**This comment is similar to the comment made above. Low DIC concentration in the ground waters of SW region could also be due to high dilution and different soil characteristics (lithology). Though we have discussed the influence of ground water DIC concentrations on the export flux and yield of DIC from the Indian peninsular rivers, we could not quantify the ground water DIC flux to the coastal waters (Arabian Sea and Bay of Bengal) and their contribution to surface DIC export flux as we have not determined the ground water exchange rates, which is not within the scope of the present study.**

*Reviewers comment*
Line 469: soil organic carbon content….what is the source for this data?

*Author's response*
**Soil organic carbon data has taken from *Kishwan et al.,* 2009 and *Sreenivas et al.,* 2016. This will be mentioned clearly in the revised manuscript.**

*Reviewers comment*
Suggested Literature:
1. Bhavya, P.S., Sanjeev Kumar, Gupta, G.V.M., Sudharma, K.V., Sudheesh, V. (2018). Spatio-temporal variation in d13CDIC of a tropical eutrophic estuary (Cochin estuary, India) and adjacent Arabian Sea. *Continental Shelf Research*, 153, 75-85, doi: 10.1016/j.csr.2017.12.006.

2. Gupta, G.V.M., Sarma, V.V.S.S., Robin, R.S., Raman, A.V., Jai Kumar, M., Rakesh, M. and Subramanian, B.R (2008). Influence of net ecosystem metabolism in transferring riverine

organic carbon to atmospheric CO2 in a tropical coastal lagoon (Chilka Lake, India). *Biogeochemistry,* 87: 265-285, doi:10.1007/s10533-008-9183-x

*Author's response*
**These two reference will be cited as you suggested for Cochin estuary and Chilka lake respectively.**

---

## Author Response (AR1)

The paper by Krishna et al. provides a very large and almost complete data set on DIC concentrations in the estuaries of Indian rivers and the δ13CDIC. The authors use the data to calculate DIC discharge to the Arabian Sea and Bay of Bengal and DIC yields from the catchments. It is a very important and valuable data set collected in 2011 and 2014 and should be published and made available also in the light of ongoing global change affecting the amount and sources of riverine DIC discharged to the ocean basins.

**Thank you very much.**

In the methods section the authors state that they have done multiple sampling in the estuaries. The standard deviation should be given in Figure 2.

**Standard deviation has been given in Figure 2 as you suggested**

In the methods the authors indicate that they have DO and chl-a data which they present only in a summarized form in Figures of the paper. These also need to be made available in an attachment.

**This data will be made available**

The interpretation of the data is rather convoluted and therefore difficult to follow. It needs restructuring and would benefit from one or more tables.

**The discussion part has been restructured following your suggestions given below**

Furthermore, an introduction to the use of $\delta^{13}C_{DIC}$ as an indicator of DIC source is missing. Such information is given in lines 320-337. This could fit into the introduction.

**Description on the use of $\delta^{13}C_{DIC}$ as an indicator of DIC source has been shifted into the introduction section. P. 4-5, L. 89-105.**

Generally chapters 4.1 and 4.2 may be merged if the discussion is organized differently, may be as suggested below. Now some aspects of the $\delta^{13}C_{DIC}$ results are mentioned in 4.1 and 4.2 repeats some of the earlier arguments. In order to better organize the discussion a Table would help showing the average rainfall in the four regions, the volume of discharge per m-2 and soil OC (lines 308 ff). To my mind the last paragraph of chapter 4.1. may rather be the starting point of the discussion.

**The structure of the discussion has been modified by considering your suggestions. In order to avoid repetitions, the chapters 4.1 and 4.2 are now merged and it was started with the last paragraph of the chapter 4.1, as you suggested. Average rainfall, volume of discharge and soil OC data of the four regions have been provided in a separate Table (Table 2)**

In general I would suggest to follow a clear structure in the discussion (Chapter 4.1.), discussing consecutively (for example). (Most of these points are already mentioned in lines 237-245):
- dilution effects and mixing effects with sea water (this point may well be discussed in the beginning to exclude certain samples from detailed source discussions using δ13CDIC)

- The impact of rock weathering: carbonate vs. silicates, which rock types dominate the catchment?
- The impact of soil organic matter

**Structure of the discussion chapter has been revised following your suggestions. It has been re-organized in three sub-sections (4.1.1 to 4.1.3) under the section 4.1. The first sub section 4.1.1 has been dedicated to discuss the impact of hydrological conditions (including mixing and dilution) while the influence of in-stream process were discussed in the second sub-section 4.1.2. Subsections 4.1.3 dealt with the impact of lithology and soils in the catchment. By re-organizing in this way, many repetitions have been deleted.**

In Schulte et al. I found a very good sketch of the contrasting impacts of these two mechanisms on $\delta^{13}C_{DIC}$ which may provide a helpful concept
- Primary production and respiration in the river/catchment
- possible anthropogenic impacts.
- More points..?

**A sketch showing the $\delta^{13}C_{DIC}$ range of DIC derived from different sources to rivers has been provided now (Figure 5). The influence of physical and biogeochemical processes such as photosynthesis and decomposition of organic matter on $\delta^{13}C_{DIC}$ has also been shown in this figure.**

The following collections of data could be given as Tables:
lines 225-233: Table of DIC concentrations selected rivers;
lines 390-396: Table of DIC discharge and
lines 415-418: Table of DIC yields of various regions/rivers.

**Data in lines 225-233 was provided in Table 1**
**Data in lines 390-396 was provided in Table 3**
**Data in lines 415-418 was provided in Table 3**

Detailed comments:

Line 109ff: the studied rivers are perennial so that there is most probably some discharge during the non-monsoonal months and some of the river catchments may even receive winter rains so that these sentences have to be formulated a bit differently. The term "monsoonal rivers" is OK but the discharge during the other seasons may be stated as "small".

**This paragraph has been modified and the discharge during the dry period was stated as small. P. 6, L. 134.**
**There is a small amount of discharge during the winter monsoon but will be stores in dams/reservoirs. P. 6, L.127-130.**

Lines 177/178: delete sentence

**The sentence was deleted**

Lines 238-245: this is part of an introduction.

**This part has been shifted to an introduction chapter. P. 2-3, L. 50-57.**

Lines 247 ff: the groups of rivers with high, intermediate and small discharges may be indicated.

**Discharge from all four groups of rivers was provided in Table 2.**

Lines 255-260: this part is redundant, shorten.

**This section (L. 255-260) has been modified. P. 15, L. 326-328.**

Lines 320-337: introduction, see above. This part and the entire chapter have to be carefully checked. Atmospheric CO2 has a value of -7 to -8 ‰ but dissolved in water; Line 323: this is not clear: CO2 has a $\delta13C$ of -7 to -8 ‰ but when dissolved in water it is around 0 ‰ if the main anionis HCO3-

**Apologies for the mistake. Here, we meant to say that DIC originated by dissolution of atmospheric $CO_2$ (-7 to -8‰) is close to 0‰. It was corrected now. P. 4, L. 92**

Chapter 4.4 may be substantially shortened and I would suggest to shift the discussion on the source of DIC to the earlier chapter 4.1.

**Chapter 4.4 has been revised and shortened considerably. Many of the redundancies were deleted, and the discussion on source of DIC has been deleted as it has already been discussed in the chapter 4.1.**

Lines 440ff is this contradicting the earlier discussion that groundwater is low in the SW and that this substantially is responsible for low concentrations?

**This was deleted during revision because it is a repetition. The discussion on the impact of submarine ground water discharge (SGD) on DIC has already been made in the chapter 4.1.**

Lines 450ff: the whole discussion on lithology could also be better in chapter 4.1.
May be a shorter discussion on the reason for the different yields (sediment/rock types and elevation; dams) would be sufficient.

**The impact of lithology on DIC has been discussed in the sub-section 4.1.3 of the chapter 4.1 as you suggested. Here, we mention only the role of soil organic carbon content in increasing DIC yield from the SW rivers in a short paragraph. P. 20-21, L. 458-467.**

The authors promise to send the data on request by E Mail. However, they should be made available in a data bank or as an attachment to the paper.

**Data is accessible through the website of our Institute's data centre (http://www.nio.org/iodc)**

The abstract is quite good and may be retained even after changing the discussion. Likewise, changes of the summary would be also rather small after a revision.

**Thank you. Summary will be refined.**

**REVIEWER 2**

The manuscript by Moturi et al. presents an extensive database on the DIC fluxes to the Indian Ocean from the monsoonal rivers in the Indian subcontinent. It is clearly the outcome of very hard work, which resulted in this important dataset. As such, this work is valuable, and I can imagine that if published, this dataset would be used by modelers and other researchers.

**Thank you very much.**

However, at its current form, the manuscript suffers from two essential drawbacks, which in my opinion, should be corrected before the manuscript can be published or even properly reviewed.
My first and main concern is with the quality of the presentation, namely the writing. The manuscript is heavily burdened by numerous grammatical mistakes, redundancies, and unintelligible sentences. Since English is not my native language, I am well aware of how hard it is to write in a foreign language, and therefore, I strongly urge the authors to have their manuscript edited by a native English speaker and/or by a professional editor.

**Since English is not our native language, we have a limited control on the English language and therefore there are mistakes in grammar, phrasing, syntax and connecting words in the manuscript. Most of such mistakes were corrected during the revision.**

My second concern is with the somewhat superficial interpretation of the data. The authors relay heavily on correlations to investigate the relations between different characteristics of the rivers, but correlations do not necessarily imply cause/effect relations or, as the author argue in the discussion section. Therefore, the conclusions the authors draw are rather general, and do not go much beyond the data itself. Consequently, the manuscript has more resemblance to a report, and may be more appropriate to publication as such. I would recommend the authors to consider more carefully how this dataset can advance what we have already learned from previous works. At its present form, it is very hard to evaluate the scientific contribution of this work, and therefore, I recommend that this discussion paper be withdrawn, and perhaps submitted ab-initio after it has been thoroughly edited and revised.

**We have used correlations between various parameters to explain the impact and its statistical significance of different processes within the catchment and rivers on the export and yield of DIC from the Indian monsoonal rivers to the northern Indian Ocean. Though the correlations do not necessarily imply cause/effect relations, as you said, we performed these correlations only to understand the major processes controlling the DIC export fluxes from Indian monsoonal rivers. We have not based only on correlations alone but we described different sources and processes responsible for distribution and export fluxes of DIC from the India monsoonal rivers.**

General comments:

Grammar: The text is laden with grammatical errors. The first sentence in the abstract, for example, is flawed. So are lines 19-22, 65-68, 177-178 and many more.

**The first sentence in the abstract has been corrected. P. 1, L. 10-11**
**Lines 19-22 have been revised. P. 1, L. 19-22**
**Lines 65-68 were corrected. P. 3, L. 70-73.**
**Lines 177-178 were deleted as the Reviewer 1 suggested deleting these lines.**

The usage of connectors ("However", "Though", "Despite" etc.) is wrong throughout the text.

**Mistakes associated with English language such as grammar, phrasing, syntax and connecting words were rectified.**

Writing: Many sentences in the text are excessively long and incomprehensible (e.g. lines 30-35, lines 225-232). Reading sentences that contain more than 80 words, 11 values and more than 10 references (lines 225-232) is extremely demanding, and prevents the reader from understanding the messages that the authors try to convey.

**Long and incomprehensible sentences have been modified to obtain the clarity in conveying the message to the readers.**
**For example, the sentence in lines 30-35 has been split into two sentences. P. 2, L. 29-37.**
**Lines 225-232 have been modified and restricted to 64 words (reduced from 117 words) by providing DIC concentrations of different rivers in a table (Table 1).**
**P. 12, L. 265-269.**

Cumbersomeness and redundancies: Far too many results are incorporated in the text, instead of being presented as figures (e.g. lines 225-232, 247-250). This makes the manuscript cumbersome and turns the reading into a very demanding task.

**Many of the results mentioned in the text have been given in Tables. For example, DIC concentrations mentioned in lines 225-232 are now given in a table (Table 1).**
**Results mentioned in line 248-249, 309 and 425-428 were given in Table 2.**
**Results mentioned in lines 390-396 and 415-418 were provided in Table 3**

Some statements repeat themselves along the text (e.g. lines 176-177 and lines 216-217), making the text needlessly long.

**Lines 176-177 were deleted as the same was mentioned in the discussion.**
**P. 11, L. 254-255.**

In some parts of the manuscript, there are no references to existing figures. Instead, the authors re-cite the values (see previous comment), whereas in others, the authors refer to relationships which should have been presented as figures (e.g. lines 263-264. See also detailed comment #22).

**All figures have been cited in text. Relationships mentioned in the text have been presented as figures. Figure 4 (4a to 4f) and figure 6 (6a to 6h).**

Units: The authors report most of their DIC data in mg l-1. This unit is somewhat archaic, and unclear. To what does the "mg" refer? Bicarbonate? Carbon? The more explicit concentration units of mol l-1 or mol kg-1 are much more common in the current literature. The authors themselves use mol kg-1 in the methods section. In the same section, they use percentage to describe the accuracy. This usage of multiple units for the same parameter is needlessly confusing. I recommend reporting all the results in mol l-1 or mol kg-1.

**DIC concentrations throughout the manuscript were expressed in mg $l^{-1}$. Though it is old, we used mg $l^{-1}$ because we expressed DIC export in Tg $yr^{-1}$ and DIC yield in g $m^{-2}$ $yr^{-1}$. Further, same y-axis has been used for both DIC concentration and yield in figure 2. In the unit mg $l^{-1}$, 'mg' refers to the total dissolved inorganic carbon but not only bicarbonate. In the methods section also, units for precision have been changed mg $l^{-1}$ in order to maintain the consistency throughout the manuscript. Accuracy of the method is generally expressed as percentage of error, the units (%) used for accuracy of our method for determination of DIC remain unchanged.**

Error propagation and significant figures: In the methods sections, the authors report the analytical errors associated with their concentrations measurements. However, they do not propagate these errors to the DIC fluxes. In addition, the authors report too many significant figure compared to the error they report.

**The errors associated with flux and yield estimations have been provided in figure 2 in the form of error bars. Significant figure was followed.**

Figures and missing figures: Figures 1 and 2 are clear and informative. The rest of the figures are correlations, and could be presented in one or two panels. For some reason the authors did not include figures for some of the correlations they describe in the text. I cannot understand why.

**All the correlations mentioned in the text have been presented as figures in two panels (figures 4 and 6).**

Specific comments: (please note that I did not include all the grammatical errors in the text).

Line 10: change to "rivers are an/a important/significance source of: : :"

**The sentence has been modified as 'Rivers an important source of ...' P. 1, L. 10**

Line 19-22: revise this sentence. The usage of connectors is grammatically wrong. Use "enriched" instead of "caused the enrichment". Also, the "stable isotopic composition" cannot be "enriched". Use either "enriched in 13C" or "increase _13CDIC values"

**The sentence has been revised. It has been split into two sentences. 'caused the enrichment' has been changed to 'enriched $\delta^{13}C_{DIC}$' P. 1, L. 19-22.**

Line 25: The sentence is grammatically wrong

**The sentence has been modified. P. 1, L. 25-26.**

Line 30: define "yield of DIC"

**'yield of DIC' has been defined. P. 2, L. 29.**

Lines 30-35: This sentence is too long and unintelligible

**This sentence has been split into two sentences. P. 2, L. 29-37.**

Line 56: "The Mississippi river"

**It was deleted during revision.**

Line 65: How do the fresh water discharge, and suspended sediment load relate to the fluvial carbon fluxes?

**Freshwater discharge significantly influences the fluvial carbon fluxes to estuaries and coastal region as it scours terrestrial carbon from rocks and soils. However, suspended load was deleted to convey the message more clearly. P. 3, L. 70-73.**

Line 67: The sentence is grammatically wrong

**It was corrected. P. 3, L. 70-73**

Line 71: change "estimating" to "estimations"

**The sentence was modified. P. 4, L. 76-77.**

Lines 73-76: Most of the rivers mentioned in this paragraph are located between 30°S - 30°N. So why do the authors claim for ": : :paucity of data" (line 72) for this region?

**Here, we meant to say that many of the medium rivers from this region were not included in the global DIC estimations due to the paucity of data. The rivers mentioned in lines 73-79 (Mississippi, Congo, Changjiang and Pearl) are only a few of the large rivers in the world. However, to obtain the clarity, the sentence has been modified. P. 4, L. 76-77.**

Lines 76-81: The sentence is grammatically wrong

**The sentence has been modified. P. 4, L. 81-86.**

Line 82: The phrasing of this sentence is awkward, consider revising

**The sentence has been modified. P. 5, L. 106-107.**

Line 154: The units here are different from the units used in the text. Please be consistent. It is advised to use mol kg-1 throughout the text

Unit 'mg $l^{-1}$' has been used to express DIC concentrations throughout the text as explained earlier. It has been changed here also. P. 8, L. 174

Line 155: change "Scripts" to "Scripps"

**Sorry for the mistake. It has been corrected to 'Scripps' P. 8, L. 175**

Line 157: If the CRM from Andrew Dickson lab is used, 0.3 % equals approximately ±6 µmol l-1. This is considerably larger than the precision the authors report in line 154. This error should be propagated along with other sources of error, to calculate the error on the flux estimations

**0.2 to 0.3% is the error associated with the accuracy of DIC determination while the value given in line 154 is the precision of the method. However, the precision has also been changed to mg $l^{-1}$ to maintain the consistency. As mentioned earlier, errors associated with determination of DIC concentrations and standard deviations of the mean values were propagated to DIC export flux and yield calculations. These errors have been shown in figure 2 in the form of error bars.**

Line 177-178: This sentence's phrasing is awkward, consider rephrasing

**This sentence has been deleted, as suggested by the Reviewer 1**

Line 179: If the error is in the second significant figure, it makes no sense to report 4 significant figure. Change 30.86±1.23 $^o$C to 31±1 $^o$C (and throughout the rest of the manuscript)

**Results have been presented up to the significant figure for all the parameters.**

Line 205: remove the comma after "The estuaries"

**'Comma' has been removed. P. 11, L. 243**

Lines 216-217: This was already stated in lines 176-177.

**The sentence in lines 176-177 has been deleted to avoid the repetition because it was mentioned in the discussion. P. 11, L. 254-255.**

Line 232: These values were already mentioned in line 224.

**The value has been deleted. P. 12, L. 269**

Line 236: I suggest that the mean values be added to figure 1 or to figure 2

**Mean DIC concentration in the each region was provided in figure 2 as you suggested**

Lines 247-250: There are way too many values in this sentence.

**These values were deleted from the text and were provided in a table (Table 2)**

Lines 255-277: The authors describe 4 correlations here. None of them is shown in a figure, whereas other correlations are. Why did the authors chose not to show there correlations in figures? Since the readers cannot see the fit the authors used, there is no point in mentioningthe (very poor) $R^2$ values.

**Figures were provided for all the correlations mentioned in the text. They have been presented in two panels (Figure 4 and 6).**

Line 328: The sentence is grammatically wrong

**The sentence has been corrected. P. 5, L. 97-99.**

Line 501: Add the NIO number or remove this sentence

**Contribution number will be added only after the manuscript has been accepted for publication (during galley proof correction)**

**REVIEWER 3**

This paper presents a hard work from the extensive field coverage of 27 Indian monsoonal estuaries twice during the discharge period of two different years. In the growing concern of climate change when many of the biophysical and biogeochemical models are suffering from the lack of data sets from the tropical rivers, I am sure this paper once published will significantly fill that gap and heavily used by many researchers. However, the manuscript requires to provide clarity and corrections on certain issues before it is published.

**Thank you very much**

The DIC concentrations and fluxes are influenced by the rainfall variability among the four regions, the discussion will be benefited if it starts with this information.

**Yes. Concentration and fluxes of riverine DIC are strongly influenced by the variability in rainfall over the catchment of the river (region). We have started the discussion with the rainfall variability over the four regions (NE, SE, SW and NW) of India and its impact on distribution of DIC concentrations in Indian estuaries. P. 12-13, L. 278-293**

From Figure 1, it is apparent that many of the east flowing rivers, especially in the central and southern regions, are sourced from the western catchments but none in the vice versa direction. This is important and highlighted because high rainfall SW regions have less discharge and DIC fluxes but much of this rainfall might be sourcing the less rain fed SE rivers and contribute to high DIC fluxes. I strongly suggest the authors to include a Table of all rivers sampled (grouped into four regions) with details of their size-class (large and medium), catchment size, length of the river, soil organic carbon, discharge rate, mean DIC concentration, export flux, yield, etc. for better utilizing the hard work of this study by scientific community.

**A table (Table 2) has been provided in which the rivers (grouped into four regions) were given along with their characteristics such as the catchment area, annual mean**

**discharge, soil OC and precipitation. Mean (±SD) values of concentrations, export flux and yield of DIC from each group was also provided.**

Many of the statements are repeated throughout the manuscript which makes it length, for example, parts of section 4.2 and 4.4 carry some common information. Restructuring of discussion by appropriately merging relevant subsections will improve the focus and clarity.

**The discussion chapter has been completely re-organized by merging the sections 4.1 and 4.2 (as suggested by the Reviewer 1) to avoid repetitions. Under this section, the impacts of hydrological conditions, in-stream processes, catchment lithology and soil organic carbon on DIC concentrations has been discussed, and this information has not been repeated. The repeated information in the section 4.4 was deleted, and this section has been considerably shortened.**

Number of figures can also be minimized, for example, merge figs.4 & 5 and 6 &7.

**As suggested by the Reviewer 2, figures for all the correlations mentioned in the text were given now. However, to minimize the number of figures, as you suggested, all the correlation figures were merged and given in two panels, i.e., figures 4 (4a-4f) and 6 (6a-6h).**

The manuscript requires thorough editing for English grammar for better reading.

**Many of the mistakes in English language have been corrected and the quality of English language has been improved.**

**Specific Comments:**

Line 43: delete 'about'.

**'about' has been deleted. P. 2, L. 46.**

Line 54: it is an obvious statement, delete.

**The sentence has been deleted**

Lines 55-57: how much increase? Specify 'Mississippi river'.

**As per *Ren et al.,* (2015) the total increase in DIC export throughout the 21st century from the Mississippi River to Gulf of Mexico would be over 90% due to the combined effect of climate-related changes along with rising atmospheric $CO_2$ . However, this sentence was modified during revision. P. 3, L. 60-62.**

Lines 76-81: include carbon studies from Gupta et al. (2008) in the Chilka lake, a brackish water estuarine system. Also, include Bhavya et al. (2018) for Cochin estuary.

**Studies on Chilka lake by *Gupta et al.* (2008) and Cochin estuary by *Bhavya et al.* (2018) have been included. P. 4, L. 81-88.**

Lines 81-82: Carbon export fluxes from the Chilka lake (Gupta et al., 2008) and Cochin estuary (Gupta et al., 2009) on east and west coast of India respectively were earlier reported.

**This sentence was added. P. 4, L. 86-88.**

Lines 95-102 & 120-124: Too big sentences.

**Big sentences in lines 95-102 and 120-124 have been modified. They have been split into two sentences to obtain the clarity. P. 5-6, L. 117-124 and P. 6-7 L. 142-146, respectively.**

Lines 132-134: Year 2011 was a normal monsoon year but 2014 was an El-Nino year.

**The mean values of normal monsoon and weak monsoon (El Nino) provides better mean concentrations rather than the mean of two normal monsoon years (expected to be higher side than long term mean) or two weak monsoon years (expected to be lower than long term mean). Therefore, field sampling in this study was conducted one during the normal monsoon year and the other during the weak monsoon year**

Please comment or speculate the variability in light of having used discharge data of earlier years from the published literature. Authors may refer to Indian Annual Rainfall Statistics reports available online at www.imd.gov.in.

**Since the aim of our study is to estimate the export fluxes of DIC, the long term mean values of discharge provide relatively better estimates than the discharge values in a year or two due to strong inter-annual variability in discharge (as it depends on the strength of the monsoon). IMD provides data on annual rainfall statistics over the Indian subcontinent but not the volume of discharge from each river which is crucial for flux quantification. However, we have used the IMD rainfall statistics over the Indian subcontinent to discuss the spatial variability in DIC concentrations in the Indian estuaries.**

Lines 134-137: These are contradicting the statements made at lines 130-131.

**Lines 130-131 mean to say that it is from starting point (origin) to ending point (estuary) of the river, i.e. entire length of the river; whereas Line 134-137 means that it is the length of the estuary (upper and lower estuaries) but not the entire length of the river. However, these sentences have been modified to obtain the clarity. P. 7, L. 149-154.**

Line 139: replace was with 'were'.

**Sorry for the mistake. 'was' was replaced with 'were' P. 7, L. 157**

Line 174: specify the source of catchment area.

**Source of the catchment area of rivers has been given. P. 9, L. 195-196**

Lines 185-186: give mean±SD values.

**Mean±SD values for chlorophyll-a have been provided. P. 10, L. 222-223**

Lines 207-208: delete 'by the Indian monsoonal rivers'.

**'by the Indian monsoonal rivers' was deleted. P. 11, L. 245**

Lines 216-17: repeated statement

**The sentence in lines 216-217 is a repeated one with the lines 176-177. Since the sentence in lines 176-177 was deleted, the sentence in lines 216-217 retained.**

Lines 250-254: Provide full details in a Table for better usage of this work by many researchers.

**These details have been provided in Table 2.**

Line 256: Include Gupta et al., 2008 for Chilka lake. Bhavya et al. 2016 covers only dry season (postmonsoon), replace it with Bhavya et al. 2018 for all seasons.

**Gupta et al. (2008) has been included and Bhavya et al. (2016) has been replaced with Bhavya et al. (2018). P. 15, L. 326-328**

Lines 260-262: Rather relationship with TOC (DOC+POC) is better.

**A positive relationship between DOC and DIC indicates that addition of DIC by heterotrophic decomposition of POC which gives both DOC and DIC during heterotrophic transformation. Because of this reason we have taken only DOC instead of TOC**

Lines 282-286: It seems this ground water regional variation is following the variability of DIC in the regional estuaries. Does this mean the cause factors for DIC variation are also applicable for its variation in the ground water? Please make a statement on this.

**Since ground waters are one of the important sources of DIC in estuaries, it is possible that ground water DIC concentrations will have significant impact on DIC concentrations in estuaries. However, due to the influence of other factors such as hydrology, lithology and environmental characteristics of the catchment on DIC concentrations in estuaries, it is very difficult to make a statement that ground water is the only cause factor for variability of DIC concentrations in estuaries. However, we discussed the influence of ground water (SGD) on DIC concentrations. P. 14, L. 315-324.**

Line 285: provide units for all the values.

**These values have been deleted from the text and provided in a table (Table 2) as suggested by the other Reviewer.**

Lines 286-289: Grammatically sentence not correct.

**The sentence has been modified. P. 14, L. 320-322.**

Lines 304-307: Please comment, if not speculate, on whether these soil characteristics are limited only to surface or extended to the vertical strata as well, which can give an insight into whether the source of low DIC in these surface and ground waters are same or different.

**We considered the characteristics (lithology) of only surface rocks/soils in the catchment area of the river/region from available soil maps of India. We have not discussed the vertical strata of the rocks/soils. It is possible that vertical strata of the rocks could have influenced the low DIC concentrations in ground waters of the SW region. However, we have not focussed on the reasons for spatial variability in ground water DIC concentration as it is not the scope of this study.**

Lines 310-312: Weathering rates may be high due to highest precipitation but DIC flux from the weathering of lateritic soils to the SW estuaries (refer lines 304-307) could have been far lower than other regions.

**DIC concentrations and export flux are far lower in the SW region than the other regions and it could be due to dilution, SGD with low DIC concentrations and the dominance of lateritic soils in the catchment of SW rivers. However, the dense rainfall over the SW region increases the scouring of DIC from soils and therefore causes elevated yield of DIC (DIC export normalized by the river catchment area) from SW rivers. It has been discussed in the section 4.1 (4.1.1 to 4.1.3)**

Lines 316-318: better integrate these with statements made at lines 365-471 and attribute to intense precipitation, presence of less weathering lateritic soils and soil organic carbon.

**A statement has been given on the sources/controlling processes of DIC in the Indian estuaries. P. 17, L. 375-377**

Lines 325-330: both the statements correspond to the weathering but the contribution of d13CDIC values were differently reported. Pls check.

**Though, both the statements correspond to the weathering of silicate and carbonate rocks by carbonic acid, the resulted $\delta^{13}C$ of DIC is different because it is based on the formation mechanism of carbonic acid. Carbonic acid can be formed by dissolution of soil $CO_2$ or atmospheric $CO_2$, both of which have different $\delta^{13}C$ values. Weathering of silicate and carbonate rocks by soil $CO_2$ yield the $\delta^{13}C_{DIC}$ values of -17 to 21‰ and -7 to -8‰, respectively. Respective $\delta^{13}C_{DIC}$ values would be -7 to -8‰ and -3 to -4‰ if weathering occurs by dissolution of atmospheric $CO_2$. Therefore, the $\delta^{13}C$ of DIC is dependent on the source for formation of carbonic acid, i.e. soil $CO_2$ or atmospheric $CO_2$.**

Lines 352-355: Repetition of statements at lines 326-330 but with clarity here. Avoid repetition.

**Lines 352-355 have been deleted during the revision**

Lines 395-396: Are these discharge per day or year?

**These discharges are per year. It has been mentioned in the text. P. 18, L. 415-416**

Line 401: Relatively higher export fluxes…….. compared to what?

**The sentence has been modified. P. 19, L. 420-424.**

Line 405: When combined…..with DIC export flux?

**The total fluvial dissolved carbon flux is the sum of DIC and DOC. It has been clearly mentioned in the text. P. 19, L. 426-429**

Lines 424-425: SW region is having highest rainfall but lowest discharge rate from smallest catchment area. If so, large amount of rainfall might be happening over the non-catchment area. What would be the fate of this? Please discuss on the possibility of its seeping into the ground water and its contribution of DIC flux to the SW coast of India, its relativity with respect to surface flux?

**The density of rainfall is high in the SW than the other regions of India. However, lower river discharge from the SW rivers is mainly due to the small catchment area of the SW rivers than the other peninsular rivers. We have discussed the influence of ground water on the concentrations and export flux of DIC from the Indian monsoonal rivers. However, we could not quantify the ground water DIC flux to the coastal waters (SW and SE coast of India) and their contribution to surface DIC export flux because we have not determined the ground water exchange rates, which is not within the scope of the present study.**

Lines 432-433: include -ve sign for the r2 values for having the negative relationships.

**Negative sign '-' has been given for the $r^2$ value, if the relationship is inverse.**

Lines 441-442: Reference to the comment for lines 424-425. Does low DIC concentration in the ground water of SW region is also due to high dilution rate and possible lateritic soil strata? Please comment on what would be the ground water discharge rate and its associated DIC export flux to the SW coastal AS compared to the other regions.

**This comment is similar to the one mentioned above. Low DIC concentration in the ground waters of the SW region could also be due to high dilution and different soil characteristics (lithology). Though we have discussed possible influence of ground water exchange on the concentrations and export flux of DIC from the Indian monsoonal rivers, we could not quantify the ground water DIC flux to the coastal waters (Arabian Sea and Bay of Bengal) and their contribution to surface DIC export flux because we have not determined the ground water exchange rates, which is not within the scope of the present study.**

Line 469: soil organic carbon content….what is the source for this data?

[revised manuscript text omitted]

Fig. 2

[Figure]

Fig. 3:

[Figure]

[Figure]

Fig. 4:

[Figure]

Fig. 5:

[Figure]

Fig. 6:

Fig. 7:

[Figure]

[Figure]

Fig. 8:

---

## Author Response (AR2)

Major comments

Method for DIC fluxes calculation
L200 "Total export flux of DIC… multiplying the mean concentration of DIC in an estuary
with the annual discharge. Spatial variability of DIC in estuaries was minimized to a large
extend by collecting samples from head to mouth of the estuary" Estuarine saline samples
contain a fraction of seawater; their DIC is thus a mixture of riverine DIC and marine DIC,
with proportions depending to the value of salinity. The concentrations of DIC in these
estuarine samples are affected by dilution with seawater and cannot be multiplied by the
freshwater discharge to derive a flux to the ocean. According to all textbooks on estuarine
hydrology and chemistry, the DIC concentration at the zero salinity end-member must be
used. Concentration versus salinity plots can be used to obtain an "apparent zero end-
member" (AZE). The method used here is underestimating the real DIC flux when freshwater
DIC cc is higher than seawater DIC cc (most cases in these indian rivers), and, inversely,
overestimating the flux when river DIC < ocean DIC.

**The samples collected at stations of only near zero salinities (riverine end members)
were used in this study and saline samples (salinity >2) were excluded. In a very few
medium estuaries where samples were not collected at near zero salinities, the riverine
end member was calculated from the DIC versus salinity plot, as you suggested. As we
have considered only riverine end members for this study, the text was modified
accordingly and confined only to rivers.  The export fluxes of DIC are estimated by
multiplying DIC concentrations at riverine end member with the discharge of the river.**

The use of the term "estuary" all through the MS in often inappropriate. At many places in
the MS, in the abstract, introduction and discussion, you mix the two notions of river and
estuary. You use the word "estuary" when it should be "river". See below

**Since we have considered only the riverine end member samples for this study, we have
used only 'river' throughout the manuscript, i.e., in the abstract, introduction and
discussion.**

Interpretation of 13C DIC
In general, the discussion of the 13C DIC signal is confusing and too superficial. It must be
strengthen by more quantitative analysis that allows to evaluate the respective contribution of
processes under some specific conditions. You mention weathering of silicate/carbonate and
"storage" in dams as major processes. However the relationship between DIC cc and 13C and
lithology is only verbal. You must relate DIC concentrations and 13C-DIC signature with
lithology and the proportion of carbonate versus silicate rocks on the respective watershed.

**The discussion on the 13C DIC was changed completely and re-written.  It was given in
a separate section 4.2. Since the 13C DIC measured in the riverine end member samples
is a mixture of different carbonate species (HCO3, CO3 and CO2) which are largely**

**dependent on pH and temperature. Further, in-stream physical and biological processes alter the 13C of source. In order to separate the influence of different in-stream processes on 13C of DIC source, we tried to approximate the 13C of DIC source using two different graphical model techniques, Keeling plot and Miller-Tans plot and interpreted the outcome of the models for DIC sources. Deviations of the measured 13C DIC from the approximated 13C of DIC source were attributed to the influence of in-stream processes. Further, to filter the influence of pH and DIC speciation on bulk13C DIC, we approximated the 13C of CO2 using a set of isotopic fractionation enrichment factors across the DIC speciation. With this approach, we found that 13C DIC in most (~75%) of the Indian rivers is influenced by heterotrophic decomposition of organic matter, whereas in the rivers from southeast region of India, it is controlled by autotrophic production. Overall, we tried to explain the 13C DIC in a more quantitative manner.**

Include lithology in the map in figure 1 and provide in a table the proportion of carbonate and silicate rocks in each watershed. The table should also include classical information for each watershed (discharge, drainage, presence of dams, eventually population density). All information necessary for the quantitative information of the data should be included.

**Lithology was included in the map (Fig. 1), as you suggested. A table (Table 1) with all the necessary information such as catchment area, discharge, length of the river, number of dams on the river and population density in catchment area of the river was provided. However, we could not get the information on proportions of silicate and carbonate rocks in each watershed, and thus not provided in the table.**

Sections 4.1.2 and 4.1.3 on the origin of DIC is extremely speculative and sometimes repetitive. We must know what are major processes and minor processes. Pollution by megacities is ignored although it is probably a major driving force on DIC in these rivers.

**As mentioned above, the discussion on 13C DIC was changed completely and it was re-written to present the results in a more quantitative way. The major processes influencing the 13C DIC were identified using graphical model approach (Keeling and Miller-Tans plots). To filter the influence of pH and DIC speciation on the bulk 13C DIC, we have approximated the 13C of CO2. Influence of pollution on DIC was also included now as it is also important to be considered.**

Detailed comments

Abstract
L14: spatial and/or temporal variability?

**It is spatial variability. It was mentioned the text.**

L17: "exchange of groundwater with low DIC" this is speculation because groundwater are supposed to have high DIC concentrations

**Yes. DIC concentrations are higher in ground waters than the DIC in rivers/estuaries. Spatial variability of ground water DIC showed the lowest mean values in the SW region than the other regions of India. We mean to say here that exchange of ground waters with relatively lower DIC in this region (but not lower than the estuarine DIC) than the other regions. However, the sentence was modified during revision.**

L21: "intrusion of marine waters": intrusion of seawater occurs in all estuaries (not only east-flowing estuaries), and are more or less important depending of sampling strategy and salinity of the samples. Comparison between systems can be done only on the basis of the zero salinity end-member.

**DIC concentrations at stations of only near zero salinities in each river (riverine end member) were considered for this study. Since the saline samples were not included in this study, the sentence was modified accordingly.**

L27-28: "9.4% of the Asian rivers" unintelligible sentence

**Since the Indian rivers are part of the Asian rivers, this part of the sentence was deleted.**

Intro
L44 "DIC is the major constituent of carbon species" where, please rephrase

**It was rephrased as "Dissolved inorganic carbon (DIC) is one of the major constituent of carbon species in rivers'.**

L46 "33-400 Tg of DIC" do you mean 330-400 Tg? Please proofread you MS to remove such typos before re-submitting

**Sorry for the mistake, it was 330 only but not 33. However, it was given more precisely in the revised text.**

L50: should be "rivers" instead of "estuary" here

**Since only the riverine end members were used for this study, 'river' was used throughout the manuscript.**

L51-55: rephrase and specify what process is more important quantitatively

**During re-structuring the manuscript, the major processes controlling DIC concentrations in the Indian monsoonal rivers were discussed in the 'discussion' chapter. Hence, it was deleted from here.**

L62-65: this is too general, how does human alterations of riverine DIC relate with findings of your study?

**These two sentences were deleted as our findings are not related with human alterations. However, the influence of population density on DIC concentrations in the Indian monsoonal rivers was discussed in the section 4.1 and shown in Fig. 4g**

L70-88: this is too general, and not focussed in relation with the present study. Please rewrite

**Entire 'Introduction' chapter was re-written. During this process, this section (L.70-88) was deleted.**

L88 "their sources" of DIC?

**Yes. It is 'their sources of DIC'. However, this sentence was deleted from the text as it is related to the estuarine export of DIC (including saline samples) to the coastal ocean.**

L90 aquatic systems

**Corrected to 'aquatic systems'**

L91-93 "due to distinct isotopic composition of different sources" not only as 13C-DIC is also affected by fractionation by internal processes such as aquatic primary production and gas exchange. The figure 5 should be used here (as new figure 2) to support this statement

**The section was modified as ".Though the $\delta^{13}C$ of DIC derived from different sources is well separable (Deines et al., 1974), the isotopic fractionation by in-stream physical and biological processes alters the $\delta^{13}C$ of DIC source (Fig. 2). For example, photosynthesis and equilibration with atmospheric $CO_2$ enriches (O'Leary, 1988; Finlay, 2004; Parker et al., 2005, 2010) while the heterotrophic decomposition of organic matter and photo-oxidation of dissolved organic carbon depletes the $\delta^{13}C$ of DIC (Opsahl and Zepp, 2001; Finlay, 2003; Waldron et al., 2007; Vahatalo and Wetzel, 2008) (Fig. 2)".**

**The figure 5 was given as figure 2 to explain various in-stream processes influencing the 13C of DIC.**

L100: "despite distinct isotopic composition of DIC is expected for different sources, the identification of DIC sources is still challenging due to isotopic fractionation associated with complex mixture of sources and processes such as photosynthesis" awkward sentence. Mixture does not fractionate. All through the MS you must prioritize what is more important among the different processes that impact 13C-DIC

**The sentence was shifted to section 4.2 (sources of DIC). It was modified as "Though, the $\delta^{13}C_{DIC}$ is a promising tool to decipher the sources of DIC, its interpretation for**

**source material identification in rivers is still challenging because multiple physical and biological processes within the rivers significantly alter the $\delta^{13}$C of DIC source. The influence of major in-stream processes on the $\delta^{13}$C$_{DIC}$ must be separated before interpreting the results for major sources of DIC, failing which leads to erroneous conclusions. In order to identify and separate DIC sources, we used here two different graphical mixing model techniques, Keeling plot (Keeling, 1958; Pataki et al., 2003) and Miller-Tans plots (Miller and Tans, 2003). These models approximate the hypothetical $\delta^{13}$C of source material as an intercept (in Keeling plot) and slope (in Miller-Tans plot) of the least square linear regression equations (Pataki et al., 2003; Campeau et al., 2017). The deviations from the approximated $\delta^{13}$C of source can be interpreted to the influence of in-stream processes".**

Study area
We need a table with all necessary information about each watershed

**A table (Table 1) was provided with all necessary information of each watershed.**

L121_124 Not sure information on upwelling in the Arabian sea is necessary here

**Since the rivers draining into the Arabian Sea were also covered in this study, the information on the Arabian was also provided along with the Bay of Bengal.**

L138 rivers instead of "estuaries"?

**It was changed to 'rivers' as you suggested**

L147 when talking about discharge, better refer to "rivers" rather than "estuaries"

**The term 'rivers' was used throughout the manuscript**
Sample collection
We need to know about salinity of your samples. Comparing data with high salinity from an estuary with data with low salinity in another estuary is meaningless. Salinity must appear somewhere in the MS, and what should be compared is DIC and 13C-DIC at the zero salinity end-member.

**Salinity samples were not included in this study. Hence, discussion on DIC and 13C-DIC was confined to freshwater regions.**
L200: use the AZE DIC concentration to calculate fluxes

**Mean of the DIC concentrations measured at stations of near zero salinity in each river was used for DIC export flux estimations. In some cases, where measurements at near zero salinity were not conducted, apparent zero end member was estimated from the relationship of DIC with salinity, as you suggested.**

L214: here you mix different information (river composition and dilution with seawater) and the fact that salinity is high in an estuary is due to your sampling strategy. All interpretation should refer to river end-member, and should not include different proportion of dilution with seawater

**DIC concentrations at near zero salinity were used for this study. No saline samples were included in this study and thus no contribution of DIC from seawater. Hence, comparison, discussion and interpretations made in this study were confined only to the river end member.**

Section 3.3: re-calculate DIC fluxes using the zero salinity end member

**DIC fluxes were re-calculated using the measured DIC concentrations at near zero salinity (river end member). For few medium rivers, where samples were not collected at near zero salinity stations, the riverine end member was obtained from the DIC versus salinity plot, as you suggested.**

L258 "no vertical salinity stratification was observed in all estuaries sampled… Mandovi estuary" awkward sentence

**The sentence was deleted during revision as the saline samples were not included in this study. Hence, stratification in estuaries is irrelevant here.**

L273 "among the estuaries sampled along the indian coast, the SW estuaries are characterized by lower mean concentrations of DIC than the SE, NE…" awkward sentence, please rephrase

**The sentence was modified during the revision.** '**Distribution of DIC in the Indian monsoonal rivers showed large spatial variability, with the lowest values in rivers from the SW region of India (Fig. 3a)'.**

L279 "the SW region of India receive the highest amount of precipitation during the SW moosoon than the SE, NE and NW regions of India" awkward sentence, please rephrase

**The sentence was corrected and rephrased as 'The spatial distribution of rainfall over the Indian subcontinent ([www.imd.gov.in](http://www.imd.gov.in)) shows that the SW region receives the highest annual rainfall (~3000 mm) than the rest of India (Soman and Kumar, 1990). '**

L286 "About three times higher catchment area normalized discharge might have diluted DIC concentration in the SW estuaries" not clear, please rewrite

**It was modified to obtain clarity. 'In order to understand the influence of the density of rainfall on DIC in rivers, we normalized the volume of discharge from the river with its catchment area. . The catchment area normalized volume of discharge was found to be much higher in rivers from the SW region (1.71 m$^3$ m$^{-2}$) than the rivers from SE (0.17 m$^3$ m$^{-2}$), NE (0.6 m$^3$ m$^{-2}$) and NW (0.32m$^3$ m$^{-2}$) regions of India. About three times higher catchment area normalized discharge might have diluted DIC concentrations in the rivers of the former region.**

L289 rivers instead of "estuaries"

**It was changed to rivers. The sentence was re-written as 'A strong exponential decrease in DIC concentrations with increasing rainfall over the catchment ($r^2$= 0.72, p<0.001; Fig. 4a) also suggests that DIC concentration in the Indian rivers are strongly influenced by density of precipitation over the catchment.'**

L290-293 not sure dilution will alter the 13C-DIC. You mention "residence time of soil water" but this is not clear: what process will deplete 13C DIC ? Be more precise when referring to Amiotte Suchet 1999

**The sentence was deleted during the revision. Discussion on 13C –DIC was provided in a separate section 4.2 (sources of DIC) to maintain the focus of the study.**

L296: you mention here "lower catchment area" but the info on catchment areas is missing in the MS

**Catchment area of each river was provided in the Table 1**

L300-304: rewrite and provide more quantitative evidence for the occurrence of in stream processes. Here the interpretation of the data is only verbal. What instream process are you referring to?

**It was re-written and some part of the paragraph was deleted. It was re-written as 'Since the contribution of DIC from in-stream processes, such as decomposition of organic matter, has been demonstrated to increase along the course of fluvial network (Hotchkiss et al., 2015), possibly due to increase in the residence time of water (Catalan et al., 2016), the lowest DIC concentrations found in rivers from the SW region may also, at least partly, be due to their small size. Fairly good positive correlation between DIC concentrations and length of the rivers ($r^2$=0.38, p<0.01) also support this argument.'**

L309-311. Comparing low salinity 13C signature in a given estuary with high salinity 13C-DIC signature in another estuary makes no sense. The fact that 13C-DIC correlates with salinity in a given estuary is a truism. These types of statements must be avoided in all the MS

**The section on 'mixing with seawater' was deleted as the saline samples were not included in the present study.**

L320 "exchange of SW estuaries with…" unintelligible sentence: how can you exchange en estuary?  All the literature converges to the conclusion that groundwater have high DIC concentrations, so your statement here is pure speculation

**The sentence was deleted as the estuarine (saline) samples were not included in this study**

L328 "DIC input from the dissolution of atmospheric CO2 can be ruled out". Isotopic equilibration of DIC with the atmosphere occurs even if waters are oversaturated with respect to the atmosphere.

**Isotopic exchange is true, but here, we meant to say that the addition of DIC to rivers through the dissolution of atmospheric $CO_2$ is more unlikely.  However, the sentence was modified to avoid confusion.  It was modified as 'Since the Indian monsoonal estuaries have been reported to be a source of $CO_2$ to the atmosphere during the discharge period due to heterotrophic decomposition of organic matter (Sarma et al., 2001, 2011, 2012; Gupta et al., 2008, 2009; Bhavya et al., 2018),  the DIC input from the dissolution of atmospheric $CO_2$ may be unlikely.  On the other hand, organic matter decomposition is expected to add significant amount of DIC as enhanced bacterial respiration rates were reported during this period (Sarma et al., 2011; 2012'.**

L331-336 this is speculation : the fact that 13C DIC correlates positively with DOC does not necessarily mean that oxidation of organic matter predominates. High 13C DIC can be due to fractionation by phytoplankton and high DOC to the exudation/hydrolysis of phytoplanktonic cells. The correlation can also be indirect, High DOC and high 13C DIC coming from the nature of soils and rocks (or pollution) on the watersheds, or can also be due to the dilution of seawater in the samples. The highest value of DOC in fig 4e at about 13 mg L-1 may also come from contamination by sewage. See also below my comments on figure 4.

**Yes. Uptake of DIC by phytoplankton enriches 13C DIC and exudation of phytoplankton cells increases DOC concentration, leading to a positive relationship between the two. Yet, it is also due to carried forward signatures from the catchment as you suggested.  Hence the section was re-written as follows:**

**On the other hand, organic matter decomposition is expected to add significant amount of DIC as enhanced bacterial respiration rates were reported during this period (Sarma et al., 2011; 2012).  In contrast, significant negative correlation between chlorophyll-*a* and DIC ($r^2$=-0.44, p<0.01; Fig. 4c), except few SE rivers where elevated phytoplankton biomass (Chl-a: >5 mg m$^{-3}$) was recorded, suggesting that autotrophic removal of DIC is also significant in the Indian monsoonal rivers during the study period. A significant positive relationship was observed between the $\delta^{13}C_{DIC}$ and Chl-a ($r^2$=0.49; p<0.01; Fig. 4d), supporting this argument because preferential uptake of $^{12}C$ than $^{13}C$ during photosynthesis leaves the residual DIC enriched in $^{13}C$.  On the other hand, $\delta^{13}C_{DIC}$ showed significant positive correlation with DO saturation ($r^2$=0.50, p<0.01; Fig. 4e) (depleted $\delta^{13}C_{DIC}$ values at more under saturation of DO) and DOC concentrations ($r^2$=0.43, p<0.01; Fig. 4f) as was observed in the Xi river (Zou et al., 2016).  Altogether, enriched $\delta^{13}C_{DIC}$ are associated with higher DOC, less under saturation of DO and higher phytoplankton biomass (Chl-a) while the depleted $\delta^{13}C_{DIC}$ are associated more under saturation of DO and less DOC. This suggests that both autotrophic removal and heterotrophic addition control DIC in the Indian rivers during the discharge period, with a considerable spatial variability.  However, influence of these processes on DIC concentrations is difficult to separate with this bulk $\delta^{13}C_{DIC}$ data set, as the $\delta^{13}C_{DIC}$ in rivers is also influenced by pollution, catchment lithology and outgassing of $CO_2$ (Shin et**

**al., 2011; Brunet et al., 2005; Bouvillion et al., 2009; Zeng et al., 2011; Tamooh et al., 2013). Excluding Sabarmati and Mahisagar rivers, DIC concentrations showed fairly good linear relationship with population density over the catchment of the river ($r^2$=0.41, p<0.01; Fig. 4g), suggesting that considerable influence of pollution from the mega cities and industries on DIC in the Indian rivers.**

L338 "is distinctly enriched than that…" unintelligible language

**The sentence was deleted during revision**

L339-342: the contribution of C4 plant material is speculative. Many other processes may enrich DIC in 13C such as carbonate weathering (which also increases alkalinity), gas exchange and mixing with seawater DIC.

**This section was modified as 'These results indicated that DIC in the Indian rivers is largely contributed by chemical weathering of carbonate and silicate minerals by soil $CO_2$ (-10 to -9‰). Deviations of the measured $\delta^{13}C_{DIC}$ (-13.0 to -1.4‰) from that of the approximated $\delta^{13}C$ of DIC source (-3.0 to -2.0‰) and $\delta^{13}C$ of $CO_2$ (-10.7‰) could be due to the influence of in-stream process. In more than 75% of the Indian rivers sampld, the deviation from the $\delta^{13}C$ of DIC source is towards negative side (depletion) ($\delta^{13}C_{DIC}$ < -3.0‰), suggesting that heterotrophic decomposition of organic matter is the dominant process controlling DIC in these rivers. While, no (or very little) deviation was observed only in rivers from the SE region of India (mean $\delta^{13}C_{DIC}$: -3.1‰) could be due to the competition between autotrophy, degassing and heterotrophy as these processes influences the $\delta^{13}C_{DIC}$ in opposite directions (Fig. 2); the former two processes causes enrichment while the latter depletes $\delta^{13}C_{DIC.}$. Relatively higher phytoplankton biomass (mean Chl-a: 4.6 mg m$^{-3}$) and less unsaturation of DO (98.7%) was observed in these rivers compared to the mean of the rest of the Indian rivers (2.4 mg m$^{-3}$ and 87.5% respectively), suggesting that autotrophy is one of the dominant processes controlling DIC in rivers from the SE region of India. Total number of dams on the rivers from this (SE) region (mean 155, Table 1) is not significantly higher from that of the mean of total number of dams on the Indian rivers sampled (mean 135) , suggesting that degassing due to storage of water may not be the dominant process responsible for enrichment in $\delta^{13}C_{DIC}$ of these rivers".**

We need some quantitative analysis here: what are the contributions of silicate and carbonate rocks on the respective watersheds, and does 13C DIC correlate with alkalinity? High alkalinity is an indication of carbonate weathering

**Yes. 13C DIC is correlated with alkalinity and the same was given in figures (Fig. 4h). Since the discussion on 13C-DIC was changed completely, this paragraph was modified during the revision.**

L326-377. All the section 4.1.2 is extremely speculative, discussion on processes are only verbal and not quantitative. All the possible processes are mentioned but without giving more emphasis on one process compared to another. Entire section must be rethought and re-written.

**It was completely re-written as mentioned in above responses to your comments.**

L360 "we could not able to evaluate" please proofread your MS before submitting!

**It was deleted during the revision**

L362 "the same is correlated with DO saturation and found significant…" revise language "confirming that biological processes enriched 13C DIC in the Indian estuaries. CO2 outgassing due to heterotrophic decomposition…. In reservoirs/dams…" This is extremely confusing and speculative. In fact you mention all possible processes but you don't know from your analysis which one is most important. In addition "biological activity" is too general and what most probably makes the 13C become enriched is primary production (autotrophy) and not heterotrophy.
Pollution by megacities is ignored although it is probably a major driving force on DIC in these rivers.

**The sentence was deleted, and the entire section was re-written. Influence of pollution on DIC in the Indian rivers was also discussed. During re-structuring of the manuscript, DIC concentrations and 13C-DIC were separated in to two sections, i.e., 4.1 and 4.2, to maintain the focus of our study.**

L388 your are repeating the same statements as before

**Repeated statement was deleted**

L395 "…consumption of atmospheric/soil CO2 through silicate weathering is lower by 2 times…" is this a rule that applies everywhere? Rephrase

**The sentence was modified to obtain clarity. Lateritic soils, which are poor in lime and silicate, occupied the catchment of the rivers in the SW region of India. Relatively lower chemical weathering rates of the lateritic than the non-lateritic soils could be one of the reasons for the observed lower DIC concentration the rivers from SW region of India.**

End of section 4.1.3 is too long and repetitive

**The discussion of isotopes (end of the section 4.1.3) was deleted as it is a repetition. In order to avoid the repetitions on $\delta^{13}C$ of DIC, the entire discussion on this topic has been given in a separate section, 4.2.**

L427 80% of DIC and 20% of DOC (what about POC?) is different from other rivers in the world and not "consistent with earlier reports elsewhere"

**Here, we meat to say the fluvial dissolved (not particulate) carbon fluxes. However, the sentence was modified to avoid confusion. Contributions of DIC and DOC to dissolved carbon fluxes vary from region to region. For example, DIC dominates, as was observed in this study, in the British rivers and high altitude Swedish rivers while DOC**

**dominates in the low land Swedish rivers. It is not uniform over world.**

L435 "similar those found…" revise language

**The sentence was revised. The yield of DIC found in this study (mean 8.7±5.2 g m$^{-2}$ yr$^{-1}$) is close to those found in rivers from the tropical region of Asia, but significantly higher than those reported from tropical region of the American and African continents (Huang et al., 2012).**

L451: you repeating the same statements

**As the statement was mentioned previously, it was deleted here to avoid repetition.**

L465: "significant correlation between soil organic carbon and DIC yield … confirms that strong influence of soil and DIC yield…" a correlation does not necessarily prove a process, correlation can be indirect. However, this is probably one interesting finding that deserves more discussion: soil OC oxidation will release DIC in the form of excess CO2, but it wil also enhance weathering and the export of DIC in the form of alkalinity. Deserve more discussion

**Yes. Correlation does not necessarily prove a process, but we used correlations to understand the major processes controlling DIC in the Indian rivers. Soil OC was found to be higher in the SW region compared to the SE, NE and NW regions of India. It might have increased the yield of DIC from rivers of the former region as decomposition of soil OC releases CO2 leading to formation of acidic conditions which subsequently increases the dissolution of carbonate minerals and chemical weathering of carbonate and silicate rocks.**

L489 (and abstract) "storage of water in dams" you must consider the probable occurrence of phytoplanktonic production that would decrease DIC and increase 13C DIC.

**Phytoplankton production in rivers was found to be the dominant processes controlling the 13C DIC in rivers from the southeast region of India. Whereas, heterotrophic decomposition was found to be dominant in the other Indian rivers studied. However, the average number of dams on the rivers from SE region is close to the average number of dams on the Indian rivers, suggesting that influence of water storage in dams on 13C DIC in these rivers is minimal. It could be due to competition between different processes that enrich (autotrophic production and degassing) and depletes (heterotrophic decomposition of organic matter) the 13C DIC in dams.**

Owing to the population density in India, DIC is probably largely affected by the release of sewage waters

**Yes. DIC concentrations in Indian rivers are affected by the pollution, Population density in the catchment area of the rivers showed linear relationship with DIC concentrations in river as shown in Fig. 4g.**

Fig1: please show lithology and calculate the respective proportion of carbonate and silicate in each watershed

**Distribution of soils was shown on the map in Fig. 1. However, we could not get the information on respective proportions of carbonate and silicate in each watershed.**

Fig2 and 3 can be combined. Please place the west on the left and the east on the right as in figure 1. Reading would be improve by showing DIC yields on a seprated panel

**Fig.2 and 3 were combined and given in one panel as Fig. 3. Concentrations, export flux and yield of DIC were shown separately (Fig. 3a,b,c respectively) for better reading. West-flowing rivers were placed on the left and east-flowing rivers were shown on the right, as you suggested.**

Figs 4 and 6 are very confusing, number of data points are different from one panel to another, avoid mixing different information in such figure. Prepare one figure with a limited number of variable on the X axis. It is not clear why fig 4a, 6g and 6 h have only 4 data points: insert these ones in a different figure.

**Some of the figures were removed and some were modified. All the figures were given in one panel (Fig. 4 a-g) to obtain clarity.**

What do ovals represent is not mentioned in the captions

**Ovals represent the outliers and were not considered in the regression equations.**

In fig 4 you show 13C DIC versus salinity, and 13C dic versus DOC. What about DOC and DIC versus salinity? Salinity plots should appear in a separate figure in order to calculate zero salinity end-members

**Since DIC concentrations at stations of only near zero salinities were considered for this study, figures with salinity were deleted.**

Fig5 could appear before when mentioning the processes controlling 13C DIC in rivers in the introduction
**Figure 5 was given as figure 2 while explaining the in-stream processes influencing 13C DIC in rivers**

Table 1 is useless; what is the meaning of comparing DIC concentrations in world rivers with Indian estuaries that include saline samples?

**This table was deleted**

Table 2 should show the information for individual rivers, not only regions. Quantitative information on the lithology in the watershed should be included and used for quantitative analysis of the data

**A table (Table 1) was provided with all necessary data of each river including the catchment area, discharge, length of the river, number of dams along the course of the river and, population density and annual mean precipitation over the catchment**.

[revised manuscript text omitted]
 dissolution of soil $CO_2$ produced by the decomposition of terrestrial $C_3$ plant organic matter (~ -22‰, O'Leary, 1988; Campeau et al.,

2017; Fig. 5), suggesting that contribution of DIC from dissolution of soil $CO_2$ is of less significance. However, the observed range of $\delta^{13}C_{DIC}$ is close to the range of $\delta^{13}C_{DIC}$ derived from weathering of silicate and carbonate rocks by atmospheric $CO_2$ (-8 to -7‰ and -4 to -3‰

respectively) and soil $CO_2$ (-10 to -9‰ and -21 to -17‰, repectively) (Solomon and Cerling,

1987), suggesting the major fraction of DIC is of geogenic origin in general and weathering of carbonate mineral in particular.

[revised manuscript text omitted]

SE

Fig.1:

[Figure]

NW
(30.3±8.9 mg l$^{-1}$)

NE
(19.5±6.2 mg l$^{-1}$)

SE
(36.3±6.3 mg l$^{-1}$)

SW
(6.6±2.1 mg l$^{-1}$)

◄———Bay of Bengal———►

◄———Arabian Sea———►

Fig. 2

[Figure]

Fig. 3:

[Figure]

[Figure]

[Figure]

[Figure]

[Figure]

[Figure]

Fig. 4

Fig. 5

[Figure]

[Figure]

[Figure]

Fig. 6:

| S. No. | Mean DIC conc. (mg l⁻¹) | River | Reference |
|---|---|---|---|
| 1 | 23 | Ganga-Brahmaputra | Singh et al., 2005 |
| 2 | 22 | Hooghly | Samanta et al., 2015 |
| 3 | 15 | Mahanadi | Pattanaik et al., 2017 |
| 4 | 6-21 | York river estuary | Raymond and Bauer, 2000 |
| 5 | 28 | Yangtze river | Cai et al., 2008 |
| 6 | 4 - 43 | British rivers | Jarvie et al., 2017 |
| 7 | 18 - 22 | Seri, central Japan | Ishikawa et al., 2015 |
| 8 | 9 - 30 | Red river, Vietnam | Quynh et al., 2016 |
| 9 | 18 - 46 | Xi river, southwest China | Zou, 2016 |
| 10 | 37 - 66 | rivers draining into the Gulf of Trieste | Tamse et al., 2014 |
| 11 | 10.3 | Global mean | Meybeck and Vorosmarty, 1999 |
| 12 | 12.7 | Asian rivers in tropical region | Huang et al., 2012 |
| 13 | 3.4 to 44 | Indian estuaries | Present study |

Table 1

| S. No. | Region of India | Total catchment area (M km$^2$) | Annual Discharge (km$^3$) | Mean±SD of DIC conc. (mg l$^{-1}$) | Mean±SD DIC yield (g m$^{-2}$yr$^{-1}$) | Annual DIC export flux (Tg) | Mean±SD δ$^{13}$C$_{DIC}$ (‰) | Mean (±SD) annual rainfall (mm) | Mean±SD GW DIC (mg l$^{-1}$)* | Soil OC (ton ha$^{-1}$) |
|---|---|---|---|---|---|---|---|---|---|---|
| 1 | NE | 0.53 | 276 | 19.5±6.2 | 8.6±5.7 | 4.2 | -3.5±2.8 | 1000±200 | 92±31 | 55 |
| 2 | SE | 0.45 | 102 | 36.3±6.3 | 5.8±2.3 | 3.5 | -2.7±5.2 | 400±50 | 106±56 | 20 |
| 3 | SW | 0.02 | 46 | 6.6±2.1 | 10.8±6.6 | 0.3 | -7.4±1.9 | 2500±500 | 32±19 | 101 |
| 4 | NW | 0.21 | 75 | 30.3±8.9 | 9.5±4.0 | 2.4 | -11.1±2.3 | 750±250 | 84±54 | 34 |

*data has been taken from Dr. BSK Kumar, personnel communication.

Table 2

| S. No. | Rivers | DIC export flux (Tg yr$^{-1}$) | DIC yield (gC m$^{-2}$ yr$^{-1}$) | Reference |
|---|---|---|---|---|
| 1 | World major rivers | 385 | 2.58 | Meybeck and Vorosmarty, 1999 |
| 2 | Asian rivers | 111 | 9.79 | Huang et al., 2012 |
| 3 | American rivers | 61.4 | 3.3 | Huang et al., 2012 |
| 4 | African rivers | 17.7 | 0.63 | Huang et al., 2012 |
| 5 | Rivers draining to the tropical Atlantic from South America and Africa | 53 | - | Araujo et al. 2014 |
| 6 | Tropical rivers | 210 | 3.3 | Huang et al., 2012 |
| 7 | Indian monsoonal rivers* | 10.4 | 8.7 | Present study |

Table 3

---

## Author Response (AR3)

**Associate Editor Comments – Author Responses**

L18-22 make only one sentence, no need to repeat twice "keeling plote & miller Tans.." in the abstract

**These two sentences were combined to avoid the repetition of 'Keeling plot and Millter-Tans plot'. P.1, L. 19-21.**

L21: "the similar 13C of DIC source": what DIC sources and where, all rivers? Rephrase

**The sentence was deleted during the revision.**

L22-23 not clear what you mean by "pH and DIC speciation on measured 13C-DIC"; the sentence about 13C DIC in the abstract can be more focused on main result

**This sentence was deleted to maintain the focus on main results of the study, i.e, stable isotopes of DIC and its spatial variability. P. 1, L.16-25**

L24-25 it seems strange that 13C DIC trends are not mentioned here: depleted when heterotrophic respiration and enriched in autotrophic sites?

**Spatial variability of $\delta^{13}C_{DIC}$ was given now. Relatively enriched $\delta^{13}C_{DIC}$ values were found in rivers of the southeast region in which autotrophic production is dominant. Whereas depleted $\delta^{13}C_{DIC}$ values were observed in the other monsoonal rivers where heterotrophic respiration is predominant. P.1, L 16-25.**

L38-40 you basically repeat what was said before in the abstract without giving more detailed interesting information such as : where is autotrophy favoured (and why), where is heterotrophy favoured and why?Abstract could be improved to provide more precise information, avoiding too general statements, and repetitions

**Repeated sentences were deleted and detailed information on the spatial variability of $\delta^{13}C_{DIC}$ was given. Too general statements were deleted to maintain the focus of our main results.**

L58 and L68 Raymond et al 2013 is not the appropriate reference for DIC export fluxes to coastal regions

**Raymond et al. (2008) at L 58 was deleted and Richey et al. (2002) was added. P. 2, L. 53**

**Raymond et al. (2013) at L 68 was replaced with Joesoef et al. (2017). P. 3, L. 63**

L105 it is not clear here if "equilibrium" is the right word, rather than "isotopic equilibration". If "Isotopic equilibration with the atmosphere" refers to the changes in 13C-DIC due to selective 12CO2 and 13CO2 fluxes at the water-air interface, then the process is very significant when water pCO2 is high such as in small streams. (Deirmendjan and Abril 2018 JoH, and refs therein from J. Barth group). Please update the statements here L105-110.

**The statements given at L 105-110 were modified. "Though, rivers are generally in disequilibrium with atmospheric $CO_2$ (Raymond et al., 2013) and emit $CO_2$ to atmosphere due to oversaturation (Oquist et al., 2009; Campeau et al., 2017), the isotopic equilibration between the DIC and $CO_2$ in the atmosphere significantly influences the $\delta^{13}C_{DIC}$ in rivers (Abongwa and Atekwana, 2014; Deirmendjian and Abril, 2018) due to selective fluxes of $^{12}CO_2$ and $^{13}CO_2$ at the water-air interface. Hence, the influence of biogeochemical processes within the rivers must be considered while interpreting the $\delta^{13}C_{DIC}$ results for identification of DIC sources". P. 4-5, L. 100-106**

L154 remove "rather than headwaters" no need

**"rather than headwaters" was removed. P. 6, L.150-151**

L225 and L382under-saturation, not "unsaturation"

**Sorry for the mistake. 'unsaturation' at L 225 and L382 were corrected to "under-saturation" P. 9, L. 219 and P. 15, L. 377**

L300 exchange of CO2 with the atmosphere, and not "exchange with atmospheric CO2"

**It was changed to "exchange of $CO_2$ with the atmosphere" P. 12, L. 294**

L323 "Bouillon" not "bouvillion"

**Sorry for the mistake: It was corrected to "Bouillon" P. 13, L. 317**

L354 "interpreted "as" not "to"

**The sentence was changed to "…..interpreted 'as' the influence of in-stream processes" P. 14, L. 348**

L372 "DIC … is largely contributed by" please rephrase

**The sentence was rephrased as "These results indicated that chemical weathering of carbonate and silicate minerals by soil $CO_2$ (-10 to -9‰) is the major source of DIC in the Indian rivers." P. 15, L. 366-367**

L394 "the remaining into the Arabian Sea" please rephrase

**The sentence was rephrased as "Nearly three fourth of this amount (7.8 Tg) reaches to the Bay of Bengal while the Arabian Sea receives only one fourth (2.5 Tg)".
P. 16, L. 387-388**

L397 not sure "far less" here is correct English
**"far less" was changed to "lower" P. 16, L. 391**

What does shaded area mean in table 1?

**It was explained in the caption of the table now. "Rivers located in the northern region (north of 16°N) of India were shown by the shaded (grey) area. Of these shaded rivers, Mahisagar to Narmada (west flowing rivers) and Godavari to Haldia (east flowing rivers) represent the rivers of the northwestern (NW) and northeastern (NE) regions of India respectively**. P. 30, L. 965-968

Fig 1 is of poor graphical quality, please improve

**Quality of the figure was improved. Names of all the rivers sampled are clearly visible now. The quality of labels for different types of soils was improved.**

Fig2 I could not understand why degassing at pH <6.4 and at pH > 6.4 have different effect on 13C-DIC. Degassing will make 13C-DIC converge to about zero per mil
Weathering by "carbonic acid derived from dissolution of atmospheric CO2" is something strange, and likely not significant in soils where most of the CO2 comes from soil respiration. Where do these statements come from? Please revise this figure and improve the legend

**Figure 2 was modified and the legend was improved. Weathering of carbonate and silicate rocks by carbonic acid formed by the dissolution of atmospheric CO$_2$ was deleted as it is minor in soils. Biogeochemical processes influencing the δ$^{13}$C-DIC were also modified.**

[revised manuscript text omitted]

---

## Author Response (AR4)

**Authors Response to Associate Editor Comment**

Before you MS can be published, Figure 1 must be improved:
This figure is of poor quality and some letters appear in the background (something like "mapsof").
Please redraw to remove these letters. We also need a reference for the lithological map, it can be an published report or an official web site.

**Figure 1 was re-drawn and improved the quality of the figure substantially. A reference of the lithological map was given in the caption of this figure (Figure 1). P. 29, L. 912**